# Molecularly Imprinted Polymer-Based Hybrid Materials for the Development of Optical Sensors

**DOI:** 10.3390/polym11071173

**Published:** 2019-07-11

**Authors:** Alberto Rico-Yuste, Sergio Carrasco

**Affiliations:** 1Independent Researcher, 28400 Madrid, Spain; 2Department of Organic Chemistry, Arrhenius Laboratory, Stockholm University, SE-106 91 Stockholm, Sweden

**Keywords:** molecularly imprinted polymers, hybrid materials, polymeric composites, optical sensors, MIP-based sensors

## Abstract

We report on the development of new optical sensors using molecularly imprinted polymers (MIPs) combined with different materials and explore the novel strategies followed in order to overcome some of the limitations found during the last decade in terms of performance. This review pretends to offer a general overview, mainly focused on the last 3 years, on how the new fabrication procedures enable the synthesis of hybrid materials enhancing not only the recognition ability of the polymer but the optical signal. Introduction describes MIPs as biomimetic recognition elements, their properties and applications, emphasizing on each step of the fabrication/recognition procedure. The state of the art is presented and the change in the publication trend between electrochemical and optical sensor devices is thoroughly discussed according to the new fabrication and micro/nano-structuring techniques paving the way for a new generation of MIP-based optical sensors. We want to offer the reader a different perspective based on the materials science in contrast to other overviews. Different substrates for anchoring MIPs are considered and distributed in different sections according to the dimensionality and the nature of the composite, highlighting the synergetic effect obtained as a result of merging both materials to achieve the final goal.

## 1. Introduction

### 1.1. Molecularly Imprinted Polymers

Molecularly Imprinted Polymers (MIPs) are synthetic polymeric materials showing selective molecular recognition sites created during the polymerization as a result of the addition of molecular templates of interest [1].

Initially, functional monomers (FMs) interact with the template molecule (T) in order to perform a well-stablished FM-T complex showing a defined stoichiometry (Figure 1, step 1). This step is crucial as different interactions between these two species will rule both the selectivity and the further recognition process in the final polymer [2,3]. On the one hand, FMs can interact with T through non-covalent interactions, that is, establishing H bonds, Van der Walls interactions, ionic interactions or even hydrophobic π-π stacking, depending on the nature of the organic functional groups of both molecules; however, other FM-T can be obtained based on stronger covalent interactions thus requiring harder synthetic conditions. Different co-monomers (CMs) can be further added to the pre-polymerization mixture to modify the final performance of the material and its compatibility in different media, depending on the feed ratio of hydrophilic/hydrophobic monomers used (Figure 1, step 2). Finally, a cross-linker (CL) is required to ensure the stability of the cavity around the T in order to avoid the collapse of the material after its extraction [4]. The role of this monomer seems to be crucial for the recognition process and although the vast majority of the MIPs prepared nowadays are based on the addition of large amounts of CL (in stoichiometric excess compared to both FMs and CMs), the preparation of MIP gels using lower amounts of CL have focused the attention of the scientific community working in this field to overcome some limitations concerning kinetics (diffusion rates of T to recognition sites) and/or the hydrophilicity of the material.

The addition of solvents (S) usually avoids bulk polymerizations and considerably improves the porosity of the material, being a key aspect for recognition purposes [5]. Typically, this solvent is named as porogen due to its capacity to artificially include pores within the polymeric network in the micro- and meso-scale ranges [6]. It is worth to mention the importance of its nature, that should ensure the stability of the FM-T complex and do not interfere with the monomers during the propagation step of the polymerization, according to its polarity, dispersive forces and capacity to form H bonds [7,8]. Experimentally, it has been observed that using the same solvent during the recognition step as that used for the polymerization enhances the diffusion of the template to the recognition pockets, as the polymeric matrix keeps a kind of “memory effect” regarding its swelling properties [9]. Furthermore, the use of different FM/CM/CL/S ratios eventually yields polymers in different formats as a consequence of changes in the polymerization rates [10].

Polymerization usually takes place in the presence of initiators, molecules capable of homolitically decomposing thermal or photochemically resulting in two radicals, when the polymerization reaction is based on 1,2-additions to double bonds [11] or, on the other hand, catalysts able to perform the hydrolysis of the monomers and their further polycondensation (Figure 1, step 3) [12]. Peroxides and azo-derivative compounds belong to the former family, while acids or bases are conventionally used for the latter reaction. Due to the weak stability of the FM-T complex, particularly in non-covalent approaches, other reactions such as either ionic or coordination polymerizations have not been applied for MIPs syntheses.

Polymerization enables the creation of molecular pockets defined in terms of size, shape and functional groups distribution, that remain after the template extraction (Figure 1, step 4) as a consequence of the crosslinking degree of the polymeric matrix [13]. This step will strongly depend on the nature of the FM-T complex thus requiring harsh conditions, even refluxing the material to break the covalent interactions or just milder acidic treatments to break non-covalent interactions [14]. The last step consists on the molecular recognition revealing the selectivity of the material and the efficiency of the imprinting process through different cross-reactivity experiments (Figure 1, step 5) [15]. Each approach reveals here their advantages and drawbacks [9]. In general, covalent approaches result in well-defined stoichiometric binding sites, ensuring that one template is being recognized by on site as the functional monomers are exclusively located inside the pocket. However, the conditions used for template removal usually damages the structure, decreasing the effective number of binding sites and requiring similar extreme conditions for further recognition. This problem has been partially solved in semi-covalent approaches, where the FM-T complex showing covalent bonds is polymerized and recognition is performed as in non-covalent approaches after slightly chemical modifications of the organic moieties responsible for the recognition [16]. On the other hand, the non-covalent approach requires milder conditions to remove the template but as FMs are used in excess, they are heterogeneously distributed within the polymeric matrix, particularly onto the surface of the MIP particles resulting in non-specific interactions with molecules showing complementary organic groups, including the template molecule. Nonetheless, according to the ease of synthesis and template removal, this is still the most widespread used approach for MIPs fabrication [16].

The final MIP should be able not only to discriminate between other molecules rather than the template showing structural analogies but also to show higher affinity and well-defined binding sites that can be characterized using a broad palette of techniques, such as nitrogen/mercury porosimetry or adsorption isotherms experiments using the template or structurally-related analogues [17].

Here arises the necessity for a reference material, comprised of a similar composition, fabricated under similar synthetic conditions but showing a lack of selectivity for the template molecule. This reference should give an idea of the non-specific interactions found in MIPs, allowing to discriminate between selective and non-selective binding sites during the sorption experiments. A tough discussion is established in the MIP community as originally, the reference material was obtained under the same conditions than MIPs but in the absence of the template, being called non-imprinted polymer (NIP) [18]. However, several authors have stated some difficulties when obtaining this material, either because the absence of the template results in a different reactivity of the monomers, thus yielding different materials or even the impossibility to obtain a polymer as a consequence of the change in the polymerization rates, where propagation step becomes negligible. In order to overcome this main limitation, control-imprinted polymers (CIPs) were proposed [19], consisting of the same pre-polymerization mixture used for MIPs but using a different template inducing similar reactivity in the monomers. As the combination and ratios of different FMs, CMs and CLs are countless, the final reactivity cannot be predicted *a priori*, even using computational approaches [20], thus requiring the exploration of which approach fits better for each individual composition.

### 1.2. MIP-Based Optical Sensors

MIPs appeared for the very first time on stage in 1949, where silica gels prepared in the presence of certain dyes showed a remarkable affinity for them [21]. In this work we can find the earliest definition of a MIP, where “the adsorbent is thus pictured as automatically forming pockets that fit closely enough to the foreign molecule to hold it by van der Waals’ forces, hydrogen bonds, interionic attractions and other types of intermolecular interaction.” Therefore, it is reasonable to think of MIPs as artificial analogues of biological systems showing high affinity towards substrates. However, the definition became old-fashioned with the years, being unable to cover both covalent and non-covalent approaches that were described in the early ‘70s [22,23] and ‘80s [24], respectively. The concept “imprinted polymer” erupted in the scientific community in 1989 [25], although the “imprinting” methodology for amino-acids was properly identified before by Mosbach et al. [24] and since then the number of publications has grown exponentially until 2018, where 1007 manuscripts were released, excluding books, book chapters and patents (Figure 2).

MIPs have been widely used as sorbents for solid phase extraction (MISPE) [26], stationary phases in chromatography [27,28], carriers in drug delivery and diagnosis [29,30], micro/nano-reactors for catalysis [31], supported in membranes for purification/remediation purposes [32] and recognition elements in sensor development [33]. Although initially their application was mainly focused on SPE, their use in sensor schemes has risen accordingly to the amount of publications, surpassing the former application (420 versus 393 publications in 2018, Figure 2c) and revealing the importance of these materials and their role as potential substitutes of biological recognition elements.

Interestingly, from the 420 works mentioned above, 315 concern the development of optical sensors, including electrochemiluminescence sensors (12), which means the 75% while the remaining 25% comprises pure electrochemical sensors and sensors based on other transduction mechanisms. However, if the reader turns back to the beginning of the decade, in 2010, the opposite behaviour is observed, where the relevance of electrochemical sensors (77%) was noticeably higher than those using optical transduction schemes (23%). Before diving into the reasons behind this dramatic change of the trends, several aspects concerning the use of MIPs as recognition elements in the development of sensors should be summarized. Compared to biological recognition elements, MIPs [34,35,36,37,38,39,40]: (1) are more stable under different chemical, thermal and mechanical conditions; (2) show high performance in the presence of aqueous/organic mixtures or even pure organic solvents; (3) can be considered as “low-cost” materials, as do not require animal experimentation, the template can be recovered and further purified after the synthesis if needed and those cases where extremely toxic and/or expensive reagents are required, templates can be substituted by cheaper and non-harmful analogues; (4) can be prepared even for those compounds for which there are no biological recognition elements available. On the other hand, they: (1) show slower binding kinetics; (2) cannot respond upon binding of non-active optical analytes; (3) should be integrated and coupled to the transducer element, which sometimes is complicated due to the polymeric nature of the material; (4) are limited to the detection of only one analyte hindering simultaneous detection.

The key aspect to differentiate between electrochemical and optical sensors is the change in the optical properties of the system during the recognition event, that should be revealed and measured. In the former sensors, to avoid some of the limitations summarized previously, MIPs can be easily embedded in dense polymeric mixtures and deposited onto electrode substrates, for example by simple drop casting or simply and directly electropolymerized enabling the electrochemical transduction [41,42,43] but special efforts should be made considering optical schemes. Although several solutions have been proposed to overcome this situation, such as analyte labelling or the use of fluorescent monomers [40], they were not significantly enough to overpass the ease of fabrication of electrochemical sensors, due to the tedious procedures of synthesis. However, new fabrication procedures adapted from plastic industry (allowing to obtain not only bulk polymers but spheres, membranes or other formats with a controlled size) [13], the use of new synthetic procedures (controlled radical polymerizations instead free radical polymerizations) [44,45] or the new micro/nano-structuration techniques available (such as self-assembly or lithography, moulding, electrospinning, etc.) [46,47,48], have partially solved these issues. Kinetics can be improved by developing core-shell structures, while the use of a second material in the hybrid composite could be used not only as a support but also as the transducer element. We believe this could explain such as increase in the scientific production of MIP-based optical sensors during the last years compared to their electrochemical counterparts.

In that sense, several reviews concerning the use of MIPs for the development of optical sensors can be found in the literature based on the transduction mechanism [43,49,50] (Figure 2d), the nature of the analyte [51], the necessity of labelling [52,53,54], the fabrication methodology [55], the type of the assay [56] or even their application for real samples [30]. However, as far as we know none of them discuss in detail the relevance and advantages using hybrid materials depending on the nature of the composite and only a few of them comment on the use of generic supports but focusing on different applications [49,57]. Thus, we want to focus the attention of the reader in the different materials where MIPs can be integrated with and the advantages that they offer from the optical sensing point of view.

## 2. Bi-Dimensional Composites

There are several techniques and approaches for the manufacture of MIP-based sensors using slides or preparing films. It is necessary to differentiate both the type of polymerization and the way in which the polymers are immobilized on the surfaces of the supports, which can actively contribute to the functionality of the sensor [58] or act as a holder for it [59]. In this section we will classify the sensors based on the support on which the MIPs are deposited, namely, metallic chips, mainly gold slides, silicon-based slides and other different kinds of electrodes.

### 2.1. Metal Chips

The use of metal chips and, in particular, gold slides was popularized from the surface plasmon resonance (SPR) technique, although they have been applied to other types of detection. In this kind of devices, it is very important to obtain a surface imprinting and a control of the thickness of the polymer layer to increase the reproducibility of the measurements. There are several techniques that allow an exhaustive control of the thickness of the polymer onto the substrate (Table 1). These polymerization techniques are not only used in this type of material but are applied to most of the hybrid materials discussed in this article.

One of the simplest approaches to prepare MIP films of controlled thickness consists of depositing the pre-polymerization mixture on a substrate and spreading it homogeneously in a step prior to its polymerization. In this sense, the spin-coating technique allows a nanometric control of the thickness of the obtained layers [60]. Dibekkaya et al. [61] modified SPR chip surfaces with an MIP for the detection of cyclic citrullinated peptide antibodies (anti-CCP), using anti-CCP/AAm (acrylamide) pre-complex for the synthesis of the polymer. By this method, a nanofilm was produced as monolayer. The sensor recognition was evaluated against an immunoglobulin M and against bovine serum albumin (BSA), showing that, due to the presence of the anti-CCP/PAAm complex, there was only recognition of the analyte. It was demonstrated the possibility of reusing the sensor up to 4 times obtaining the same recognition signal, although it is possible to observe that the recovery to base line was lost with time. Ayankojo et al. [62] prepared a hybrid organic-inorganic MIP film over a gold slide for amoxicillin detection. Hybrid materials consisting of inorganic and organic components can be formed at the nanoscale and integrated on a sensor to yield uniform thin films. This technique results in a better analytical sensitivity and selectivity due to rapid mass transport, easy access of analyte molecules to the binding sites and faster recognition. This sensor allowed analyte detection in the pico- and nanomolar concentration level, remarkable when compared to other studies on amoxicillin detection by MIP-based sensors. It was found that there was no cross-reactivity against other antibiotics, structurally similar or not. However, some competition was observed for the binding sites by ampicillin, although the signal generated was low enough to be considered as a true competition.

The thickness of the polymer layer can also be controlled by confining the pre-polymerization mixture between two flat materials and applying continuous pressure during the polymerization process. One of these materials usually acts as a substrate or support for MIP while the other can be functionalized with the template molecule. This method is known as micro-contact imprinting, resulting in a surface imprint of the polymer [63]. This approach is particularly useful for the imprinting of high molecular weight species, such as biomolecules [64] or microorganism [65]. In this case, the polymer is immobilized on the gold surface using monomers with double bonds or molecules that allow the direct bonding of the MIP during polymerization. However, the use of these techniques does not allow an exhaustive control of the thickness of the MIP film. Kidakova et al. [66] prepared an SPR sensor for BSA detection by the combined use of controlled/living radical polymerization. This technique allows the control of the composition and thickness of the MIP films by using a surface initiator (called an iniferter) that also attached the film to the surface, avoiding problems related to external factors.

In some cases, polymerization control is achieved by functionalization of the support with functional monomers or with templates that allow a superficial imprinting [67]. Suda et al. [68] functionalized a gold chip with a complex molecule formed by a cyclodextrin, cortisol as template molecule and a functional monomer with a polymerizable group. In this way they develop a competitive fluorescent sensor for cortisol determination using a fluorescent probe. The use of the complex allows an oriented polymerization where the binding sites has the recognition groups in an optimal disposition. Another technique to control the thickness of MIP films consists on the use of controlled radical polymerizations (CRPs). One of the most commonly used mechanisms is the surface-initiated atom transfer radical polymerization (SI-ATRP). This mechanism is based on a reversible transfer of halogen atoms between a halide of a “dormant” chain and a catalyst based on a transition metal on a surface [69]. Saeki et al. [70] used this technique for the recognition of glycoproteins in molecularly imprinted nanocavities. The latter also used a very interesting imprinting procedure in the case of molecules based on sugars or with remains of sugars, such as glycoproteins. It was a mixed imprint that combined boronic monomers which interact covalently with carbohydrates and other monomers that interacted non-covalently with the rest of the molecule, increasing the selectivity of the nanocavities.

It is also possible to manufacture films using electropolymerization techniques [71], which will be detailed later in Section 2.3 due to its own relevance.

Not all approaches are based on the generation of films or the use of planar gold chips. Muhammad et al. [72] described a fast SERS microprobe for detection of glucose and fructose in plant tissues (Figure 3). With this approach, microprobes were prepared on gold-coated acupuncture needles via boronate affinity controllable oriented surface imprinting with the target monosaccharide as the template molecules. The glucose or fructose molecules extracted on the microprobe were labelled with boronic acid-functionalized Raman-active silver nanoparticles and thus affinity sandwich complexes were formed on the microprobes. The sensors showed selective recognition against other sugars, recognizing only glucose or fructose depending on the imprint. The LOD was comparable to other methods already published, although it demonstrated to have a wide DR and the possibility of being used in real samples to make maps of sugar distribution in plant tissues.

A final approximation is based on the immobilization of nano MIPs (nMIPs) in the gold chips. These materials have a similar size and affinity to antibodies, allowing them to be used in an analogous way. These polymers are suitable for the recognition of biomolecules and microorganisms. There are different ways to synthesize these nMIPS—by the formation of emulsions, as described by Erdem et al. [73] for the determination of *Enterococcus faecalis;* detection and by the use of templates immobilized in solid supports, as described by Ashley et al. [74] and Altintas [75] for α-Casein and vancomycin determination, respectively, allowing their determination at the level of ng mL^−1^.

### 2.2. Silicon-Based Slides

Silicon-based materials are of great interest in the development of optical sensors. In particular, slides made of SiO_2_, both glass and quartz, allow surface modifications with silanes helping to anchor the polymer and other molecules to the surface and the functionalization of the sensor. Moreover, their transparency makes them an excellent waveguide and window for interrogation with a multitude of spectroscopic techniques [80]. Table 2 shows some examples of this kind of sensors.

It is common to functionalize the surface of the support with double bonds so that the polymer remains attached after the reaction. In the case of SiO_2_, silanes containing this functionality are widely used, such as 3-(trimethoxysilyl)-propyl-methacrylate (MPS) [81]. Chang et al. [80] described a fluorescent sensor chip for indole-3-butyric acid. Pre-polymerization mixture was poured over a quartz chip previously silanized with MPS to obtain a MIP coating. With this sensor extraction recoveries were observed to be higher than 91% at μg g^−1^ concentration level. In the cross-reactivity studies it was found that structural analogues were recognized by MIP, albeit in a lower proportion than analyte, demonstrating the influence of shape on selectivity. This chip was designed to be used in an online system as a flow cell.

In these systems it is possible to use modified gold nanoparticles such as Raman probes. Li et al. [82] prepared a biomimetic recognition-based SERS sensor for the determination of acid phosphatase. The MIP layer was synthesized through self-polymerization of dopamine on the surface of glycoprotein-immobilized polymeric skeletons, prepared with a boronic acid as functional monomer. To allow SERS detection a Raman probe was prepared by using AuNPs functionalized with phenylboronic acid so that it binds to the glycoprotein recognized by the MIP (Figure 4). This sensor made it possible to reach the lowest LODs to date and also demonstrated the high selectivity against other biomolecules present in the sample. It was possible to demonstrate the stability of the Raman probes up to 5 days and the stability of MIPs, being able to be reused up to 15 times. In addition, reproducibility at the synthetic level was evaluated by studying the response obtained from five different batches, with an RSD of less than 9.2%.

The use of nMIPs has also been extended to this type of sensors. Weber et al. [83] developed a penicillin G (PenG) sensor based on the immobilization of nMIPs, synthesized via an inverse miniemulsion polymerization technique, via azida-triple bond click chemistry to the glass surface. The resulting transducers were used as sensing layers in an optical sensor, referred to as Reflectometric Interference Spectroscopy (RIfS). RIfS is a direct optical detection method based on white light interference at thin layers. The interaction of molecules with the sensitive layer is monitored by changes of the optical thickness or, in other words, changes in the refractive index revealed the recognition event. It was only proven that there was no cross-reactivity against its synthetic precursors but not against other antibiotics. The LOD reached was around mM, low enough for its application in fermentation processes. Good chip to chip reproducibility was assessed with recovery rates of three different transducers in the range of 70–120%.

Finally, Seniutinas et al. [84] work on the development of versatile SERS sensing based on black silicon (b-Si). This material with gold or silver metal coating has been shown to be an extremely effective substrate for SERS. Two different sensing modalities were evaluated. One using a MIP layer on Au-coated b-Si, for selective sensing of two closely related varieties of tetracycline. In another example, the use of a hydrophobic coating helped to concentrate the analyte adsorbed on AuNPs. It was found that the second approach allowed better results since the analyte was concentrated in a single area of the sensor rather than being spread throughout the membrane. However, the cross-reactivity was only checked with respect to an analogue and no data were provided on the LOD reached, demonstrating its usefulness but not its application.

### 2.3. Other Conductive Substrates

Electrodes based on metals, semiconductors and carbon derivatives have gained great importance in recent years in the development of sensors based on MIPs. Although their use has always been relegated to the development of electrochemical sensors it is possible to use them as optical sensors due to electrochemiluminescence (ECL) [85], a kind of luminescence in which radiative emission is obtained from an electrochemical reaction in solution. This method requires no irradiation and problems such as scattering or light source instability are avoided. Moreover, ECL affords the electrochemical regeneration of some reactants at the electrode surface and emission in ECL occurs near the electrode surface, which affords finer control [86]. It is worth to state that in this type of sensors a light-emitting species are necessary by means of the participation in an electrochemical reaction. The most common are luminol and Ru complexes, such as *tris*(2,2′-bipyridine)ruthenium(ii) complex, [Ru(bpy)_3_]^2+^ [87], although other elements such as quantum dots (QDs) [88] or up-conversion nanoparticles (UCNPs) are currently being applied [89].

Due to the very nature of the support material, the most commonly used technique for the generation of polymer films is electropolymerization, in which the precursor mixture is deposited on partially conductive substrates, generating the MIP layer by applying an electric current. The monomers must polymerize after the application of certain voltages, although it is also common to use linear copolymers π-conjugated that interact with the template and favour electrical conductivity [90]. However, it is possible to apply other techniques for the development of sensors, such as direct deposition of the MIP on the electrode, polymerization of the material after depositing the pre-polymerization mixture on it or mixing the MIP physically with the electrode material. Table 3 shows some of these examples including the type of grafting and polymerization used.

One approach is the incorporation of the electroluminescent species into the electrode itself [91]. They prepared a [Ru(bpy)_3_]^2+^/MWCNTs/nano-TiO_2_-Nafion electrode on which the MIP selective to bisphenol A was deposited. Including MWCNTs/nano-TiO_2_ nanohybrids into the Nafion film could provide an excellent basis for effective immobilized [Ru(bpy)_3_]^2+^ and improve the electrochemical and ECL properties of the complex. This sensor achieved a low detection limit, the best to date for the analyte compare with other analytical methods and showed that there was no cross-reactivity regarding other endocrine disrupters, demonstrating its applicability in water samples.

Recently, progress has been made in the use of [Ru(bpy)_3_]^2+^ by embedding the complex in SiO_2_ nanoparticles (RUDS) in a way that avoids leaking. This material is ideal for the immobilization of the complex due to its good biocompatibility, its large specific surface area and the simplicity of modification. Lian et al. [92] developed a melamine sensor based on a glassy carbon electrode (GCE) modified with a mixture of MIP and RUDS. The sensor exhibited good selectivity in the detection of melamine and its structural analogues. In addition, post-treatment was simplified due to the oxidation of melamine, which can easily leach from the binding sites during the electrochemical reaction. Electrode-to-electrode variation was studied by melamine detection using five sensors prepared in the same batch (RSD = 2.8%) and from five batches (RSD = 6.2%), showing good reproducibility of the developed MIP sensor. Cao et al. [93] proposed a similar system using [Ru(bpy)_3_]^2+^ doped vinyl-SiO_2_ nanoparticles (Ru@ethyl-SiO_2_) and a MIP in an GCE for 17ß-Estradiol sensing. Compared to other methods this sensor yields a LOD of 5 pg L^−1^ although the dynamic range is limited. Advances in nanotechnology have made it possible to improve the development of this type of sensor, facilitating ECL emission, especially with AuNPs. These nanoparticles not only improve the transfer of electrons but also produce an ECL enhancement based on the LSPR of the AuNPs, called surface-enhanced electrochemiluminescence (SEECL) [94]. LSPR has been widely used for signal enhancement but recently, increasing attention has been paid on ECL resonance energy transfer (ECL-RET or ECRET) by the leveraging of overlapped spectra of donors emission and acceptors absorption in close proximity, likewise for Förster resonance energy transfer (FRET). Chen et al. [95] take advantage of this phenomenon by coencapsulating the donor ([Ru(bpy)_3_]^2+^)/acceptor (CdTe quantum dots, CdTe QDs) pairs into a silica nanosphere. Incorporating these nanoparticles in a GCE with a MIP allowed the development of a sensor for α-ergocryptine and ochratoxin A (OTA) with low detection limits of sub-fg mL^−1^ and broad linear ranges (fg mL^−1^ to ng mL^−1^), although at this time it is a proof of concept to be carried out in other analytes and platforms.

In order to improve their performance in terms of both recognition and luminescence, sensors have been developed combining luminescent nanoparticles with other elements. Li et al. [96] fabricated a sensor with a dual recognition system comprising a MIP and aptamers selective for lincomycin. The MIP was synthesized by electropolymerization of carbon dots (CDs)-tagged DNA aptamers combined with lincomycin and *o*-aminophenol on the AuNPs-functionalized graphene oxide (Au-GO)-modified electrode. The aptamer improved the selectivity of the sensor against other drugs, including those with a lincomycin-like structure and the CDs combined with AuNPs produce an ECRET phenomenon that improves sensitivity (LOD = 160 fM) and stability while increasing the dynamic range (pM to sub-nM).

Finally, although most of these sensors are based on ECL, some examples based on other luminescent techniques have been described elsewhere. Li et al. [97] described a fluorescence switch sensor for the detection of virginiamycin based on GO-supported CDs and MIP. The fluorescence intensity of the sensor significantly decreased due to fluorescence quenching of the GO/CDs by the analyte. This system offered several advantages, such as the high sensitivity and selectivity in the detection of antibiotics from complex matrix samples. However, the binding volume and structural rigidity of MIP in the sensor still required further improvement. Intensity was recorded by five different sensors with an RSD = 2.8%, indicating good sensor-to-sensor reproducibility. Another interesting system is the one described by Capoferri et al. [98] based on the phenomenon of electrochromism (EC), a phenomenon displayed by some materials of reversible colour change in response to an external applied potential. They developed a sensor based on ITO-IrOx NPs–MIP for chlorpyrifos determination, due to the ability of the Iridium oxide (IrOx) to turn blue−black upon oxidation and colourless upon reduction. There was a dependence between time and oxidation potential to achieve the IrOx NPs colour change and the concentration of analyte, that can be detected by the naked eye and through the use of a smartphone (Figure 5). The detection limits were in the pM range without significant cross-reactivity against other similar pesticides, making it a promising alternative for the development of sensors.

## 3. Three-Dimensional Composites

The materials discussed in this section are considered hybrid materials insofar as MIP is physically or chemically bonded to another material that gives the overall system characteristic properties apart from the selective recognition in a core-shell fashion. This section includes those materials, known as nanocomposites, based on micro- or discrete spherical (pseudo)nanoparticles. MIP layers are grown onto the surface of the seed used as core, obtaining materials of the core-coating type that show a nanometric confinement [102]. Following the same classification as above, this section is distributed according to the nature of the core, being the most important difference the dimensionality of the composite. While previously bidimensional materials were considered, here we comment on 3D structures.

### 3.1. Inorganic Cores

#### 3.1.1. Non-Magnetic Metal-Oxides

There are nanocrystals based on semiconductor oxides showing luminescent properties with potential application for the development of optical sensors (Table 4). One of these oxides is ZnO, a semiconductor material with a wide-band gap, environmental friendliness, high chemical stability and perfect biological compatibility. Zhou et al. [103] developed a sensor for tetracycline using ZnO nanorods (NRs) covered by an MIP layer. They evaluated the sensor performance comparing materials obtained in the presence or absence of CTAB as a surfactant in order to create mesoporous structures. By comparison with the non-mesoporous fluorescent sensor, the mesoporous ZnO NRs@MIP displayed a shorter response time, better selective recognition and sensitive detection of tetracycline. Compared to other similar sensors, these offered a lower LOD with a high selectivity, studied against other antibiotics of similar formula, although superior to those sensors without selectivity. In order to improve sensitivity, some authors have developed sensors based on SERS enabling lower LODs. Although metal nanoparticles have been widely used for SERS, recent investigations have revealed that several metal–semiconductor heterostructures can also generate a weak SERS signal, such as ZnO [104]. The combination of metallic nanoparticles with metal semiconductors promote the SERS effects of metallic NPs under the assistance of supporting electromagnetic enhancement from metal–semiconductor heterostructures, as set out Li et al. [105] (Figure 6). They prepared a rhodamine 6G sensor based on ZnO/Ag NRs@MIPs, which allowed LODs on the picomolar level, being selective against molecules such as crystal violet or rhodamine B. A very interesting feature of this sensor was the possibility of self-cleaning due to the photocatalytic activity of the material. Under adequate UV light irradiation, templates were completely removed and the sensor was regenerated to be used up to 5 consecutive cycles.

On the other hand, Wang et al. [106] developed a sensor based on MoO_3_, also active in SERS, covered with a MIP for the selective detection of methylene blue. In this case the material was treated with acid to generate hydroxyl groups on the surface and then functionalized with MPS to further grow the polymer on the surface. As in the previous case, the photocatalytic properties allowed the template to be removed by irradiation with UV light after the polymerization, reusing the material successively, up to 4 cycles. Although the “aim and shot” system showed good analytical characteristics and did not present cross-reactivity against crystal violet, the LOD was not well defined and was not tested on real samples, without underestimating its future prospects.

#### 3.1.2. Magnetic Metal-Oxides

Materials based on magnetic cores, usually superparamagnetic iron oxides, facilitate the manipulation of nanoparticles using a simple magnet. The greatest variability is usually found in the way the polymer is obtained and the transduction mechanism used (Table 5) [108].

The simplest system is a Fe_3_O_4_ core covered by a polymer layer. An example is the system developed by Zhang et al. [109] A resonance light scattering (RLS) sensor for the detection of Hepatitis A virus (HAV) was prepared. The recognition of the virus caused the polymer to change shape and size, varying the intensity in RLS. The negligible cross-reactivity was tested against ions, proteins and viruses that could be in the sample, besides presenting lower LOD than previous works for the determination of HAV. Another interesting fact was the verification of the null cytotoxicity of these particles, which makes possible their use in imaging and as biomarkers.

Starting from this system it is possible to make modifications improving the quality of the particles in terms of stability, such as covering them with silica before generating the polymer. Coating Fe_3_O_4_ with SiO_2_ improves the dispersion of the composite in water and increases its reusability. Li et al. [110] described a Fe_3_O_4_@SiO_2_@MIP sensor for rhodamine B detection and extraction. In this case, a fluorescent monomer based on nitrobenzoxadiazole produced a FRET response in combination with the analyte, allowing the detection of traces. While describing a low LOD (0.1 nM) they do not perform the calibration curve of the sensor nor describe the DR, which leaves the system as a proof of concept.

Other authors have described sensors based on this type of core-shell composites to which they have added another type of particles that served for the transduction of the signal. Yuan et al. [111] included Au on the surface of the ferrite for its use in an ECL system. In addition, the surface was functionalized with CDs to improve keel recognition of cinchonine. The generation of the signal was produced once the particles were incubated with the sample and immobilized in a GC electrode, resulting in the emission of the [Ru(bpy)_3_]^2+^ complex afterwards (Figure 7). The null interference of ions included in the sample and the keel detection of cinchonine were evaluated, although cross-reactivity against other molecules was not verified. The reproducibility of the sensor was examined by preparing five chips under the same conditions to detect the ECL response signal with an RSD = 3.12%. Hu et al. [112] prepared a fluorescent magnetic MIP by incorporating QDs for the determination of *N*-nitrosodiphenylamine. The QDs trapped in the polymeric matrix undergo a quenching effect upon analyte binding. This system allowed to reach a low LOD and did not present cross-reactivity against other structural analogues, demonstrating its effectiveness in both tap and sea waters. Its stability was proved after 11 measurements in 100 min although its long-term stability is not precisely defined.

#### 3.1.3. Up-Conversion Nanoparticles

Up-conversion phenomena consist of the luminescence produced when a luminophore is excited with a radiation of lower energy than that corresponding to the emitted light, being classified as an anti-Stokes process. [115] This phenomenon is observed through the continuous excitation of systems doped with transition metals or rare earths and it is considered as a non-linear optic process. The most widely used and known nanoparticles are those based on yttrium fluoride doped with other metal ions. Table 6 shows some examples of MIP-based nanocomposites that use up-conversion nanoparticles (UCNPs) as a core.

Liu et al. [116] developed a sensor for enrofloxacin based on double recognition of aptamer-MIP combined with UCNPs (NaY_0.78_F_4_:Yb_0.2_,Tm_0.02_). To this end, the aptamers were immobilized on the surface of the UCNP and a complex was formed with enrofloxacin (Figure 8). In this way a mixed recognition was achieved between the aptamer and the functional groups of the MIP. It was verified that the presence of the aptamer improved the recognition while the use of the UCNP avoided the possible photobleaching that would decrease the sensitivity of the sensor. The cross-reactivity against other fluoroquinolones (FQs) was evaluated, without any increase of the signal, being effective in a range of pH values between 7–9.

Wu et al. [117] included AgNPs to their UCNPs/MIP histamine sensor in order to create a bimodal sensor based on SERS and fluorescence. In this way the SERS signal increased directly with the analyte concentration while the fluorescence decreased. This dual detection made the system more reliable as observing the two changes confirms the unmistakable presence of the analyte. Although this method does not reach a LOD as low as other methods, it widens the DR and offers a selective recognition against other structural analogues. In addition, it is the first method for the determination of histamine based on dual detection.

Wang et al. [118] reported a sensor selective to diethylbestrol based on MIP-coated UCNPs (NaYF_4_:Yb,Er). They covered the NPs with silica and functionalized with MPS so that polymerization on the surface could be controlled. The UCNPs@SiO_2_@MIP sensor did not show cross-reactivity toward other structural analogues and although it did not present lower LOD values compared to other similar sensors, it has demonstrated its simplicity and reproducibility.

Tang et al. [119] developed a sensor for quinolones by using magnetic UCPs (NaYF_4_:Yb,Er) covered by a MIP. They used the internal visible light emitted from UCPs upon photo-excitation with 980-nm radiation to locally photopolymerize a thin MIP shell covering the magnetic UCPs. The cross-reactivity was evaluated against 10 FQs, not being considered as interferences those that presented a fluorescence variation equal to the one generated by the NIP. The LODs obtained were in the μM range and describe a multi-analyte analysis, although they do not detail how the signals related to different FQs could be discriminated.

#### 3.1.4. Quantum Dots

Quantum dots (QDs) are inorganic semiconductor nanocrystals that, after irradiation with UV light, emit in the Vis region of the spectrum at a wavelength that is a function of their size. The luminescent nature of these nanoparticles has made them indispensable for the development of optical sensors [120]. The most commonly used semiconductor QDs are cadmium selenide covered by zinc sulphide (CdSe/ZnS) [121], cadmium telluride (CdTe) [122], iron selenide (FeSe) [123] and other similar materials such as manganese-doped zinc sulphide (Mn:ZnS) [124] or some kind of perovskites (CsPbBr_3_) [125], among others. In addition, new materials have been developed such as carbon-(CDs) [126] and graphene-based (GDs) dots [127]. As a consequence of their quantum confinement, these new nanostructures exhibit characteristic optical and electrical properties that have been particularly exploited for the development of sensors. Because these materials are the most studied and applied and therefore make up the densest group, we decided to focus this overview on the most recent sensors (Table 7).

The simplest approach consists of the use of QDs as cores or trapping them in a core so that their fluorescent properties are exploited after recognition. As detailed in other sections, it is common to add components to MIP that improve selective recognition, usually by double recognition. Wan et al. [128] developed a fluorescent sensor based on CdTe QDs with doubly selective binding sites for determination of neomycin. The QDs were encapsulated in SiO_2_ and then a silica-based MIP was fabricated using a silane boronate monomer, that covalently recognizes the analyte, while the rest of the monomers allowed non-covalent recognition. It was found that there was no competition for imprinting sites by structural analogues or boronate groups by D-glucose. The LOD was comparable to other similar methods, only being surpassed by a method based on MIP-SPR, although it only required 10 μL of sample and with a simple pre-treatment based on liquid-liquid extraction and solid phase extraction. Good reproducibility was assessed by detecting neomycin with three MIP sensors prepared under the same condition (RSD = 4.5%). Geng et al. [129] described a fluorescent MIP using an aptamer as a functional monomer and CdSe as a QD core for kanamycin detection. The aptamer fixed onto the polymer matrix by a “thiol-ene” click reaction and the imprinting cavities generated a synergic effect in terms of selectivity for this analyte. The LOD reached was the lowest to date, only surpassed by an electrochemical system based on AuNPs + aptamer, without requiring a complex treatment. It was also demonstrated that there was no cross-reactivity against other structural analogues, both in terms of recognition and generation of the fluorescent signal. Five batches of MIPs were prepared to detect kanamycin, showing a good reproducibility with an RSD = 2.4%.

In recent years important efforts have been made to improve the properties of this kind of sensors by modifying or adding components like QDs. Wang et al. [130] prepared a fluorescent sensor based on CdSe/ZnS QDs sensitized with carboxylated graphene (Gra). QDs can be integrated into Gra, which functions as a two-dimensional carrier material reinforcing the sensors. The integration of Gra with MIPs improves the properties of QDs@MIPs, such as binding kinetics, selectivity, sensitivity and reliability. No cross-reactivity was observed against other structural analogues and it was also verified that there was no competition for the binding sites in the presence of potential interfering or competing species. LOD values of μg L^−1^ were reached although they do not compare them with other methods.

Another alternative to improve the sensitivity of this type of sensor is the use of ratiometric fluorescence. It involves the simultaneous measurement of fluorescence signals at two or more well resolved wavelengths followed by the calculation of their intensity ratio, which decreases the problems arising from the concentration or environmental fluctuations [131]. For this purpose, Amjadi et al. [132] proposed the use of two different QDs, CDs in the core as the reference and CdTe in the mesoporous MIP (mMIP) as the probe, for celecoxib determination (Figure 9). The LOD was low and comparable to other published methods, only surpassed by an LC-MS/MS method but with a simple and fast handling. In addition, this sensor showed high selectivity to celecoxib against not only metal ions but also amino acids and some other bioactive substances, demonstrating its applicability in biological samples. It is also possible to measure ratiometric fluorescence using QDs modified with other molecules. Yang et al. [133] proposed manufacturing a sensor using CdTe/ZnS QDs modified with ligand 8-hydroxyquinoline (HQ). Chelating Zn^2+^ ions on the surface with HQ, the nanocomposite was endowed with double emission, the one coming from the newly formed ZnQ_2_ complex and the one inherent to the CdTe QDs. In the presence of the analyte, there was a FRET phenomenon between the QD and the analyte, decreasing the intensity of one of the bands while the other remains constant. The behaviour of the sensor was tested against other food dyes, which did not produce substantial or comparable changes to those of the analyte itself.

QDs of different composition and size emit at different wavelengths. This fact can be used, apart from to measure ratiometric fluorescence, to measure simultaneously several analytes. Chullasat et al. [134] described an optosensor of dual QD fluorescence probes for the simultaneous detection of cephalexin and ceftriaxone, with CDs and CdTe QDs, respectively. With the combination of both materials, two different bands corresponding to each of the analytes/QDs were observed, allowing the simultaneous determination of these two analytes with the lowest LODs to date. However, in all cases the calibrations were performed with the same concentrations of both analytes in the sample, not showing how different concentrations would affect the bands and, therefore, the determination of the compounds. The reproducibility of the synthesis of the different composites was evaluated by preparing six different batches of each fluorescence probe with an RSD lower than 1.9%.

Work is currently underway to develop a sustainable and cleaner chemistry called Green Chemistry. This trend has also led to the manufacture of QDs as described by Shariati et al. [135] The synthesis of CDs from hydrothermal treatment of cedar and their application as cores coated with MIP layers for the fluorescent determination of phenobarbital can be considered as an example. The signal was maintained even in the presence of other compounds present in the plasma sample such as ions, amino acids and other biomolecules, confirming that there is no cross-reactivity or competition for binding sites. The LOD was comparable to other published sensors, being surpassed only by the LC-MS/MS determination, with a minimum sample treatment.

#### 3.1.5. Metal Nanoparticles

The use of metal nanoparticles as cores is based on their use as amplifiers of the optical signal generated after selective recognition. As already mentioned, a well-known phenomenon is SERS, which, by means of plasmonics, causes an increase in the Raman signal of the analyte in relation to that obtained in the absence of metal nanoparticles, considerably improving the LODs [151]. The great advantage of the combination of SERS with MIPs is that they showed a double selectivity, one coming from the selective recognition of the MIP imprinting technology and the other from the different enhanced Raman spectra for each analyte [152]. The metals that are most applied in the development of these sensors are gold and silver. Although AgNPs have a greater plasmon coupling compared to AuNPs, the formers are more unstable and give rise to sensors with a limited life time. Some sensors developed with both types of nanoparticles are described below (Table 8).

Carrasco et al. [153] described the preparation of multibranched gold−silica−molecularly imprinted polymer (bAu@mSiO_2_@MIP) core−shell nanoparticles, for the selective recognition of enrofloxacin and their application as label-free nanosensors by SERS. They evaluated different synthetic pathways for gold branch growth, silica coating and polymer synthesis. In the optimized particles (growth, coating and RAFT polymerization) they discovered that a greater enhancement of the signal was obtained, because the binding sites of the polymer were not destroyed and coincided with the hot-spots (points of greater enhancement) of the branches. No cross-reactivity to other antibiotics of the same family was observed and the lowest LODs to date were obtained for this kind of sensors.

Li et al. [154] proposed a MIP sensor based on Russian Matryoshka structured molecules for ultra-trace Tb^3+^ determination. In this case a triple recognition is described based on the formation of the Tb-Ethylenediaminetetraacetic acid (EDTA) complex (being the template molecule), the recognition by β-cyclodextrins immobilized on the AuNPs surface and the specific polymer binding sites (Figure 10). The system is based on an ECL determination in which they also propose a double amplification based on the ECRET phenomenon, which occurs between the CD/Tb-EDTA system and the [Ru(bpy)_3_]^2+^ complex and an enhancement of the ECL system itself. Although polymerization occurs over the electrode, the polymer only covers the nanoparticles. It did not show cross-reactivity against other ions and allowed to reach the lowest LODs to date for this analyte compared to other sensors and analytical methods. A well-defined sensor-to-sensor reproducibility was reached by using five different sensors prepared under similar conditions for the detection of Tb-EDTA via ECL measurement, with an RSD = 3.26%.

As already observed in the previous works, the morphology of nanoparticles greatly affects the SERS signal obtained. In fact, the synthesis of anisotropic AgNPs has aroused great interest due to their singular properties. Roy et al. [155] evaluated the difference of these properties by synthesizing spherical-, rod-, hexagonal- and flower-shaped AgNPs and by synthesizing a surrounding MIP for the determination of phenformin, testing the properties of each of the nanomaterials as a function of the shape of the core. For the synthesis of MIPs, AgNPs were functionalized with 2-bromoisobutyryl bromide in order to perform an atom transfer radical polymerization (ATRP). These particles were used for both electrochemical and photoluminescent measurements and, in general, flower-shaped nanoparticles obtained the best results. Regarding the analytical performance, the LODs were lower when a photoluminescent transduction was used, although these were slightly lower or comparable to those already described in the literature. No cross-reactivity was observed against other similar compounds, although it was only checked by electrochemical detection.

Li et al. [156] also described the synthesis of Ag@MIP particles by ATRP for the SERS determination of 2,6-dichlorophenol. However, in this case the core was modified with CdTe because the heterostructures combining metals and semiconductors increase the sensitivity in SERS by participation in the charge transfer mechanisms. The LOD was estimated experimentally, being of the order of pM and the cross-reactivity was evaluated against structurally similar compounds, generating signals much lower than those obtained with the analyte itself but higher than those obtained with the NIP.

### 3.2. Silica Cores

Silica gel cores have no functionality of their own beyond supporting the polymeric shell, with the exception of photonic MIPs as described in Section 4.2. However, they are of great interest due to their synthetic versatility and the large number of functional groups with which they can be functionalized [160]. This fact has been reflected in other nanocomposites in which the main core has been covered with silica in order to improve its properties or to allow a controlled polymerization [118]. This section shows some examples where SiO_2_ is the fundamental core of nanocomposites (Table 9).

In SiO_2_@MIP systems, the signal must be produced by the interaction of the analyte with the polymer, for which RLS is used. RLS method can detect the enhanced signals by simultaneously scanning the excitation and emission monochromators with a common spectrofluorometer. This technique is generally used with large biomolecules such as viruses and proteins. Yang et al. [161] reported a RLS sensor for the specific recognition of trace quantities of Hepatitis A Virus. This approach was the first to apply RLS to virus detection, with no cross-reactivity against other viruses. The null interference of other biomolecules and ions present in the sample was also checked and it was proved that these materials were not cytotoxic, being able to be applicable to imaging and as markers. This approach has also been used for the detection of OVA using a boronic monomer for its recognition [162]. However, in this case, the results are not as good as expected as the composition of the MIP makes it pH responsive, decreasing the specificity due to swelling and shrinkage effects.

Another widely used alternative in this type of nanomaterials is the synthesis of fluorescent polymers around the silica core. The simplest approach is to incorporate a fluorescent molecule into the polymer matrix. Wang et al. [163] developed a selective τ-fluvalinate sensor incorporating allyl fluorescein into the polymeric network, which improved the stability of the sensor that could be used at least 7 cycles. Compared to other methods, the analysis time was reduced to 12 min with no cross-reactivity compared to other structural analogues. In real samples recoveries close to 100% were obtained except at high concentrations where a decrease of around 80% was observed. On the other hand, Wagner et al. [164] synthesized their own urea-based fluorescent monomers. In this case the molecule was not only a functional monomer but also a cross-linker achieving both roles (Figure 11). The polymers prepared with this last monomer presented better analytical characteristics than those prepared with the monomer without showing the cross-linker functionality, possibly due to a better incorporation of the recognition sites in the matrix. No cross-reactivity against other similar compounds was observed and its effectiveness against other validated methods was verified. These particles were used in a microfluidic system that allowed an automatic analysis of the samples in which the NIP was used to correct the non-specific response of the particles.

Another approach is the incorporation of other particles into the core surface, such as metals that enable SERS measurements [165,166]. However, the incorporation these colloids, such as AuNPs, has also been described using fluorescent detection systems. Li et al. [167] described a selective haemoglobin sensor in which AuNPs were anchored to the silica surface with a carboxyl-amino coupling. This system was based on the phenomenon of aggregation-induced emission (AIE), thanks to which the AuNPs present a large fluorescence more sensitive and durable. It was shown that ions did not influence sensor behaviour, possibly because AuNPs were protected by MIP layers and no cross-reactivity was observed against proteins of similar size. However, it was found that the signal produced by the BSA had to be taken into consideration in the measurements. This system achieved LODs around pM although its effectiveness was not proven in real samples.

### 3.3. Other Organic and Hybrid Cores

The use of polymeric particles as cores in nanocomposites has little diffusion as they are materials that, in themselves, do not present optical properties of interest, serving only as anchor points for MIPs.

Wei et al. [173] synthesized a fluorescent MIP over a P_GMA/EDMA_ core for the detection of horseradish peroxidase (HRP). Since it is a glycoprotein, the strategy followed was the use of a fluorescent boronic acid quinoline-based monomer immobilized on the surface of the core. LOD (0.02 μM) is 10 times lower than previous fluorescence nanosensors for glycoprotein and cross-reactivity against other glycoproteins was found to be negligible. However, it was observed that in the presence of BSA the fluorescence was slightly affected, although at a competitive level the binding sites are not complementary to BSA, so the chances of fluorescence were minimal. Its reusability was verified by applying cleaning cycles and it was proven how it could be used at least 5 times without the fluorescence intensity being compromised.

Metal-Organic Frameworks (MOFs) are a particular family of porous coordination polymers showing potential voids that have been used previously for the development of both optical and electrochemical sensors [174,175,176,177]. These materials consist of metal ions or metal clusters cross-linked by organic ligands showing functional groups capable to establish interactions with the former units. Although they do not show specific recognition properties similar to those found in MIPs based on complementarity, they have been used as transducer elements in combination with other materials as a consequence of their large surface areas, enhancing the diffusion of chemical species and their eventual luminescent properties, which changes can be measured after their integration with the corresponding recognition element and related to the amount of analyte in their surrounding environment. However, it is worth to mention that MOFs show size-exclusion abilities [178] that can be combined with the coordination properties of the metals used during their synthesis to elaborate sensors for relatively small molecules [177]. Their use in combination with MIPs for optical sensing has not been fully explored yet, only a few works can be found in the literature but due to the synergetic properties of the hybrid materials based on both components, it is expected that scientific production will considerably grow during the following years. The reader is referred to the literature for further information concerning MOF@MIP-based sensors [179].

Liu et al. [180] prepared a ternary hybrid composite based on a Cr/terephthalate-MOF, MIL-101, used as the imprinting matrix, CdSe/ZnS QDs as luminescent elements and a MIP as recognition element for the quantification of pyrraline in milk powders. First, MIL-101, the MOF showing one of the largest porosity described in the literature with high thermal and chemical stabilities, was synthesized and mixed with different components of the pre-polymerization mixture, consisting of QDs previously prepared, the template, Triton-X as surfactant, APTES as the functional monomer and TEOS as the cross-linker. Ammonia was added to initiate the reverse micro-emulsion polymerization by condensation of the components. The composite was deposited on a 96-well plate and the fluorescence quenching of QDs measured in a microplate reader. Although the authors claim that the sensitivity of the sensor was improved by the combination of both MOF and QDs and the MIP enhanced the selectivity, providing a negligible cross-reactivity towards other potential interfering species, the main drawback of this work is the large time required to reach saturation of binding sites (80 min). This drawback was partially solved following a similar approach by Xu and coworkers [181]. They proposed a fluorescence sensor based on carbon dots (CDs)-decorated Zn/imidazolate-MOFs, known as CDs@ZIF-8, covered with MIP layers selective to quercetin for its analysis in Ginkgo biloba capsules used as supplement. Here, both ZIF-8 and CDs were prepared independently and further mixed, allowing the CDs to penetrate through the MOF network. Radical polymerization occurred after the addition of 4-vynilpyridine, 4-VPy as functional monomer, EGDMA as cross-linker and AIBN as initiator in a mixture DMF:MeCN, heating and stirring the slurry. Fluorescence quenching of CDs was measured by simply mixing the hybrid material in suspension with different concentrations of the analyte in a cuvette, reaching the adsorption equilibrium in 15 min. However, compared to the previous work, dynamic range was shortened (0–50 μM versus 5 μM–1 mM), probably as a consequence of the lower amount of binding sites and the higher crosslinking degree of the MIP layers. Nevertheless, the sensor developed was in agreement with current legislations and the work was fully validated by comparing the results with the well-stablished HPLC method.

## 4. Other Materials

### 4.1. Optical Fibers

Optical fibres have been used for many years in the manufacture of sensors due to their simplicity and versatility. In fact, the combination of MIPs and optical fibres has resulted in economical and useful detection systems [182]. Some examples of this kind of composite are shown in Table 10.

Cennamo et al. developed several systems based on this configuration. On the one hand, they described a multichannel optical sensor that allowed the simultaneous detection of dibenzyl disulphide (DBDS) and furfural in transformer oils. The fibres were placed in parallel with a D-shape and, on the flat side, the gold and MIP were deposited. It was demonstrated the possibility of applying both blue and red shifts for the determination of analytes, although the LOD varied depending on the selected shift and the analyte [183]. From the previous concept, they designed an optical sensor for DBDS using two optical fibres coupled through an MIP. In this case, one of the fibres was connected to the lamp and it is in which the refractive index variation is produced by the presence of the analyte, while in the other fibre part of the radiation was absorbed. In this way the errors derived from the fluctuation of the source were minimized. The LODs were similar to those already published although in this case the great advantage was the versatility and the easy manufacture. As far as cross-reactivity is concerned, only the response in the presence of furfural was evaluated and no significant signal change was observed. [184,185] On the other hand, they developed slab plasmonic platforms combined with plastic optical fibres (POFs) and MIPs for the determination of furfural. This device consisted of a holder in which was located the PMMA-Au-MIP slab waveguide, which is easily replaceable, connected to two POFs. The incident radiation and the resulting radiation were at 90 degrees, avoiding problems with the radiation from the source. The results were compared to those obtained with a classic fibre optic system, obtaining better LODs, although cross-reactivity was not evaluated. The great advantages of this system were that no polishing of the fibre is required and that the flat shape improved reproducibility and the possibility of easily changing the chip after several cycles (Figure 12) [186].

Other authors proposed the use of other elements such as metals or metal-oxides in order to improve the performance of the sensor. Shrivastav et al. [187] prepared a sensor based on LSPR and LSPR + SPR for the determination of ascorbic acid. In the former, a composite of Ag and polyaniline (PANI) was prepared over the optical fibre while for the second the fibre was covered previously with Ag. The characteristics of both systems were compared and it was verified that the sensitivity was better in the SPR + LSPR system. Cross-reactivity was evaluated against other compounds that could be present in the sample, being in all cases practically negligible. The sensor can be re-used for up to 5 cycles without significant variations in the performance and a response time of 5 sec was required. It is proposed as an alternative to determine the analyte in blood *in vivo*, although further demonstrations are not provided to support this hypothesis. Usha et al. [188] developed a system for the determination of *p*-cresol using a fibre covered with ZnO/MoS_2_ and MIP based on the lossy mode resonance (LMR) phenomenon. The addition of ZnO/MoS_2_ increased the absorption of light in the nanocomposite thereby enhanced the LMR properties and hence the sensitivity of the sensor. It was verified that there was no cross-reactivity with similar compounds and the response time was 15 s, being the less time-consuming sensor for the determination of cresol.

### 4.2. Photonic Crystals

Photonic crystals (PCs) can be defined as materials showing well-defined structures in terms of size and shape, periodically and homogeneously distributed onto different supports, at least in one direction and with different dielectric constants than the surrounding media [190,191]. This arrangement leads to the appearance of a band gap responsible for the wavelength reflection if the irradiation energy matches this value. Two main approaches have been considered to take advantage of this optical phenomenon for sensing purposes using MIPs [192]: (a) assembling MIP nanoparticles onto a substrate, measuring the change in the reflected wavelength upon template binding [193]; or (b) impregnating sacrificeable scaffolds, such as silica gel or polystyrene nanoparticles, with the pre-polymerization mixture, allowing to polymerize and finally removing the scaffold by chemical etching [194], in an approach known as inverse opal technique, being the last one the most widespread technique. Several authors distinguish between those pure MIP-based inverse opals and the works where the original molding opal template is not removed [195], although from the sensing point of view the results are similar. Interestingly, these sensors can be applied not only for quantification purposes but also for qualitative tests as identification of chemical species can be observed by the naked eye when the abovementioned band gaps match wavelengths in the visible range [196]. Table 11 shows some examples of sensors prepared by combination of photonic crystals and MIPs.

A PC MIP-based sensor for the detection of sulfaguanidine in bass and lake water samples was developed by Li et al. [197] SiO_2_ microspheres were deposited onto a glass plate by a vertical deposition method, placing the support inside a suspension of the microspheres and allowing the solvent to be evaporated (Figure 13A). A PMMA-based support was used to press the opal and the pre-polymerization mixture consisting of the template, MAA as functional monomer, EGDMA as cross-linker and AIBN as initiator was poured into the space between the two supports, permeating by capillary forces within silica spheres. After heating, the system was etched with HF to remove both the glass plate and the silica used as scaffold, thus obtaining the inverse opal structure onto the PMMA substrate (Figure 13B). After the immersion of the material in solutions containing the template during 5 min, both the intensity and the reflected wavelength bathocromically shifted were measured with a fiber optic spectrometer. The dynamic range covers 5 orders of magnitude, improving the LOD values obtained before for the same analyte using HPLC or ELISA analyses and the interference of other structurally related compounds was successfully tested, being reusable for up to 5 cycles without a significant loss of its performance. However, a full reproducibility study in terms of the material synthesis was not performed in this work. Zhang and colleagues presented a similar work for the analysis of sulfonamides in white egg samples [198]. In this case, polystyrene beads were used as scaffolds, being deposited onto glass slides following the same procedure as described above. Photonic hydrogel films were synthesized using acrylamide, acrylic acid, *N*,*N*’-methylene bisacrylamide as cross-linker and ammonium persulfate as initiator in water. The resulting composite was etched with xylene and finally analyzed using a fiber optic spectrometer. Here, a blue-shift was observed as the concentration of the analyte increased. As expected, the hydrogel decreased considerably the analysis time to 5 min but dynamic ranges were shortened down to 2 orders of magnitude, maybe because of the swelling/shrinking properties of this polymer, as revealed from the non-imprinted counterpart. Two different MIP-based PCs were prepared in this work for two different sulfonamides and they demonstrated that there is a lack of cross-reactivity between each other. However, no further selectivity studies were performed towards to other potential interfering species, LODs were not analytically identified and pH conditions should be extremely controlled by buffer addition prior to the measurements, resulting in tedious and time-consuming procedures.

A very interesting example of the interference of the template molecule used during the synthesis of MIPs is described elsewhere [199]. In this work, the antibiotic enrofloxacin is smartly replaced during the polymerization with a non-fluorescent dummy analogue, as it is well known that, eventually, a few molecules of the template remain within the polymeric matrix. Authors propose the use of curved silica photonic crystals as moldings, using the inner face of cylindrical glass bottles, comparing the performance of the imprinted material resulting from the use of this support to conventional glass slides. After piranha treatment, silica arrays were created in both supports by vertical deposition. MIP synthesis was performed by using polyethylene films pressing the colloidal array where the precursors were infiltrated, consisting of the dummy template, MAA, HEMA and EGDMA. Polymerization was photochemically initiated and after etching, the fluorescence of the PCs was measured in a spectrofluorometer. Some optical considerations are indicated, regarding the huge amount of scattered photons from the curved surfaces thus requiring the use of long-wave pass filters to avoid the background. Although the sensor was not applied for real samples, the study is fully completed with the evaluation of the sensor performance depending on the size of the silica beads used for the fabrication of the PCs, solvent, pH and time. Remarkably, authors found that their curved structures enhanced 1.68 times the fluorescence of enrofloxacin compared to the same amount of analyte in suspension.

## 5. Conclusions and Future Perspective

We have presented a new review based on MIP composites for the development of optical sensors from the point of view of the materials, their nature and properties, different from others already published focusing on the analyte, the transduction mechanism or the fabrication methodology. The growth of scientific production after the 1990s regarding the use of MIPs for sensor purposes has risen exponentially, being nowadays the application area where imprinting technology focuses more its attention, compared to classic fields such as SPE or chromatography. Interestingly, inside the sensor field, electrochemical transduction schemes have been dominating the trending until the last year, when optical sensors surpassed them. We attributed this effect to the new improvements over the main limitations that showed MIPs obtained following classical fabrication methodologies, where the format was finely controlled, that is, bulks or spheres but binding kinetics or, particularly for optical sensors, their integration with the transducer element were not convenient. Moreover, using MIPs for electrochemical transduction consisted in the easily deposition of the recognition element onto a conductive substrate, sometimes embedded in a dense mixture or after chemically functionalizing the surface. Meanwhile, measuring analytes selectively recognized by MIPs in an optical fashion typically required the labelling of the template or the monomer/s or even using fluorescent competitors in immune-like assays, that used to increase the handling of the system, leading to tedious labelling/synthetic procedures and considerably rising the expenses and total assay times. Luckily, this changed and the scientific community realized that: (1) binding kinetics were improved when core-shell composites were used, facilitating the diffusion of the species and decreasing the response times; (2) the lack of optical activity could be compensated by the use of a second material showing changes when the analyte was being recognized in the surrounding close-contact thin MIP layers; (3) coupling between two different materials was performed using new surface functionalization methods and controlled radical polymerizations, such as RAFT or ATRP. Together with these three key aspects, it is worth mentioning the novel fabrication methodologies (moulding, electropolymerization, (photo)lithography, self-assembly, among others) that we believe are behind this relevant milestone in the imprinting technology.

We have split the composites according to their dimensionality, starting from these bidimensional hybrid materials used for electrochemical purposes but showing promising results as optical sensing platforms. Typically, conductive substrates are used because of their electrochemical properties, however the use of metals such as gold thin layers have allowed optical transductions based on both SPR and SERS, where plasmonics are involved. Silicon-based substrates are able to accomplish the same role by simple deposition and immobilization of metal nanoparticles, broadening the palette of measuring techniques including fluorescence and interferometry. Either metal- or carbon-based electrodes have been widely used as supports for electrochemiluminescence or other combined and singular optical phenomena such as ECL resonance energy transfer or electrochromism. However, according to the number of publications, core-shell-based composites have proven to be the most widespread approach for optical sensing. These 3D hybrids have taken advantage of the properties of the core material, thus for example metal-oxides, up-conversion nanoparticles, carbon dots and quantum dots have been used for luminescent detections, while metal nanoparticles have been focused on SERS. Other substrates used as cores, like magnetic nanoparticles or silica nanoparticles, showing a lack of optical activity, have been applied in combination with different transduction mechanisms but using their magnetic properties facilitating the separation of the hybrid from the media or because the ease of functionalization of the support, respectively. The use of other particles as cores, such as other polymeric particles or MOF crystals have been exploited during the last years as a consequence of the large surface areas enhancing analyte diffusion but requiring the addition of either fluorescent monomers or luminescent particles retained within the pores to achieve the optical signal. Deposition and growth of MIP layers onto the surface of optical fibres have emerged as a promising methodology for the development of easy-to-functionalize optical sensors, allowing the fabrication scale-up. Finally, photonic MIPs, ordered arrays of polymer particles or holes, have demonstrated to be useful for qualitative purposes, enabling the identification of an analyte of interest by the naked eye.

To conclude, although MIP chemistry can be considered a mature field, imprinting technology is a very powerful technique under continuous evolution, feeding from relevant fields such as Organic Chemistry, Analytical Chemistry, Physical Chemistry and Materials Science while including novel tools from Polymer manufacture. We believe that the possibilities regarding the amount of materials used for the development and the fabrication of the sensors are practically countless and although nowadays QDs@MIPs seem to be the preferred composite, when new materials showing interesting optical properties, such as MOFs, are perfectly known they will be exploited in detail due to their combined synergetic properties of recognition, diffusion and transduction.

## Figures and Tables

**Figure 1 polymers-11-01173-f001:**
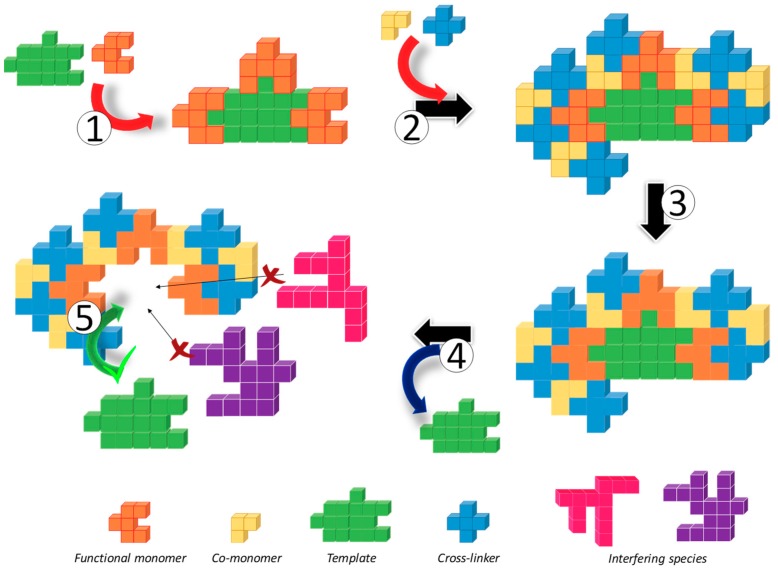
Scheme for the preparation of a Molecularly Imprinted Polymer (MIP). (1) First, template molecule (T) and functional monomers (FM) are mixed to get the complex FM-T. (2) The rest of the components of the pre-polymerization mixture are added, i.e., co-monomers (CM) and cross-linker (CL). (3) After the addition of a radical initiator (thermal or photochemical) or a catalyst (acid or base), polymerization takes place. (4) T is removed resulting in a pocket within the polymeric matrix. (5) T re-binds the selective cavity while other potential interfering species are blocked due to limitations in terms of size, shape and/or functional groups distribution.

**Figure 2 polymers-11-01173-f002:**
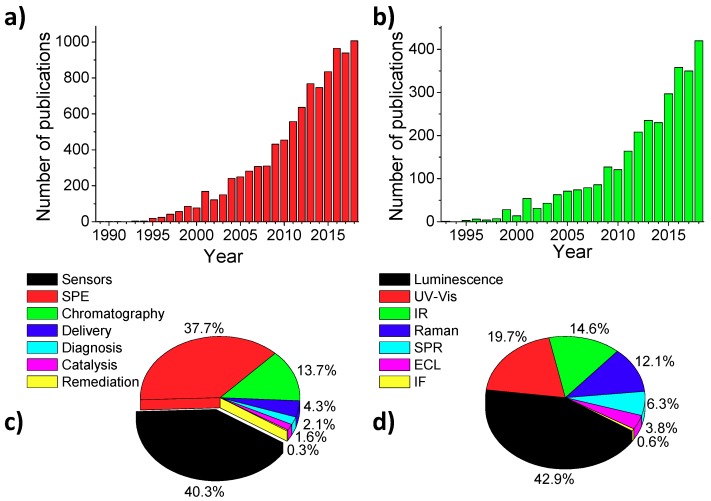
Number of publications of: (**a**) MIPs in the period 1989–2018; (**b**) MIPs used for sensor development in the period 1993–2018. (**c**) Distribution of MIP publications in different fields for the year 2018; (**d**) Optical transduction mechanisms used for the optical sensors published in 2018. SPE: solid-phase extraction; UV-Vis: ultraviolet-visible; IR: infrared; SPR: surface plasmon resonance; ECL: electrochemiluminescence; IF: interferometry. Source: Web of Science.

**Figure 3 polymers-11-01173-f003:**
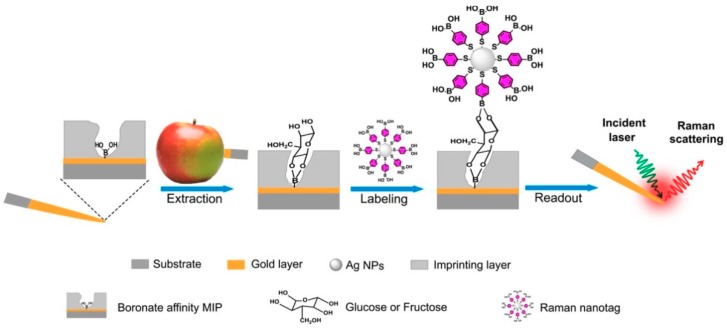
Schematic representation of microprobes prepared on gold-coated acupuncture needles via boronate affinity controllable oriented surface imprinting for glucose and fructose measurement in plant tissue by plasmonic affinity sandwich assay (PASA). Reproduced from Reference [72] with permission of Elsevier.

**Figure 4 polymers-11-01173-f004:**
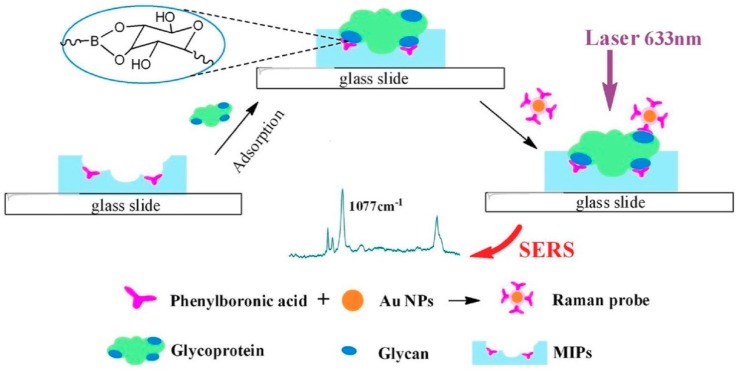
Schematic representation of the biomimetic recognition-based SERS sensor through self-polymerization of dopamine on the surface of glycoprotein-immobilized polymeric skeletons for the detection of target glycoprotein. Reproduced from Reference [82] with permission of Elsevier.

**Figure 5 polymers-11-01173-f005:**
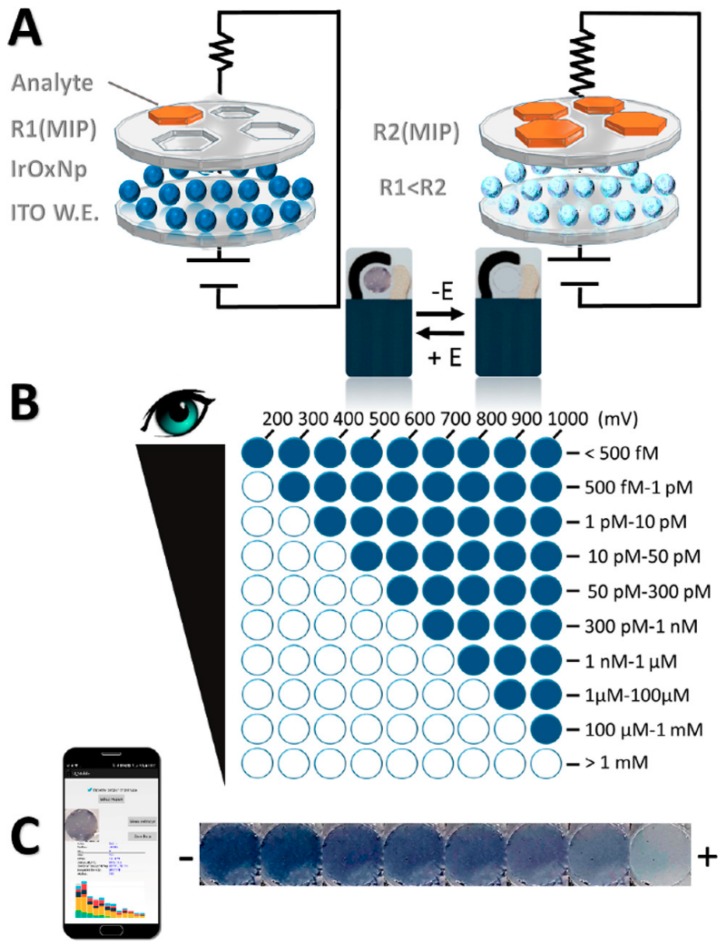
(**A**) Scheme of the MIP/IrOx NPs–ITO screen printed electrodes structure, visual IrOx NPs colour change (from blue-black to colourless) and working principle of the proposed sensor with different amounts of chlorpyrifos. (**B**) Visual detection after 10 s applying different oxidation potentials and concentration ranges detected based on the number of coloured electrodes; (**C**) change of IrOx NPs colour intensity at a fixed time and potential vs. increasing amounts of the analyte (smartphone-based detection). Reproduced from Reference [98] with permission of American Chemical Society.

**Figure 6 polymers-11-01173-f006:**
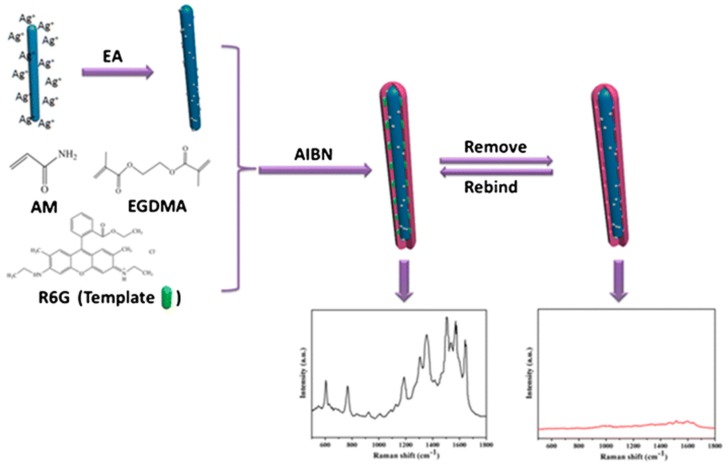
Preparation of the ZnO/Ag molecularly imprinted polymers (ZOA-MIPs) and the selective surface-enhance Raman scattering (SERS) detection of rhodamine 6G (R6G). AIBN: 2,2′-azobis(2-methylpropionitrile); AMm: acrylamide; EA: ethanolamine; EGDMA: ethylene glycol dimethacrylate. Reproduced from [105] with permission of Springer.

**Figure 7 polymers-11-01173-f007:**
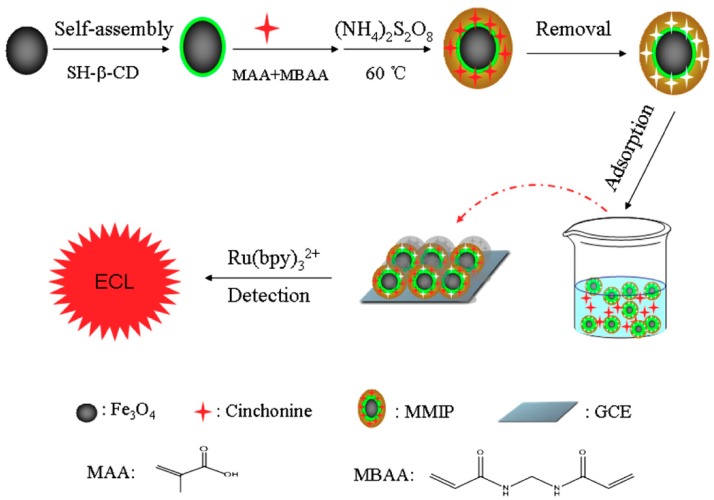
Schematic fabrication of the ECL sensor Fe_3_O_4_@Au@MIP and its use for cinchonine recognition. Reproduced from [111] with permission of Wiley.

**Figure 8 polymers-11-01173-f008:**
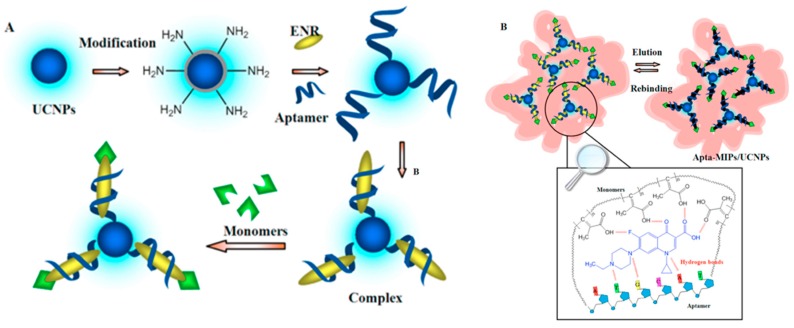
Schematic representation of the preparation (**A**) and recognition (**B**) processes for the up-conversion nanoparticles (UCNPs)@Apta-MIP selective to enrofloxacin. Reproduced from [116] with permission of Elsevier.

**Figure 9 polymers-11-01173-f009:**
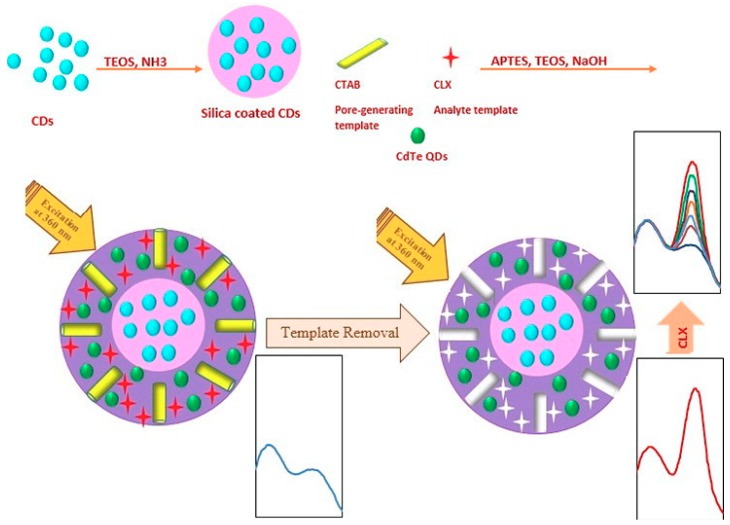
Schematic illustration of the mMIP@CDs/QDs hybrid sensor and the working principle for the detection of celecoxib. Reproduced from [132] with permission from Elsevier.

**Figure 10 polymers-11-01173-f010:**
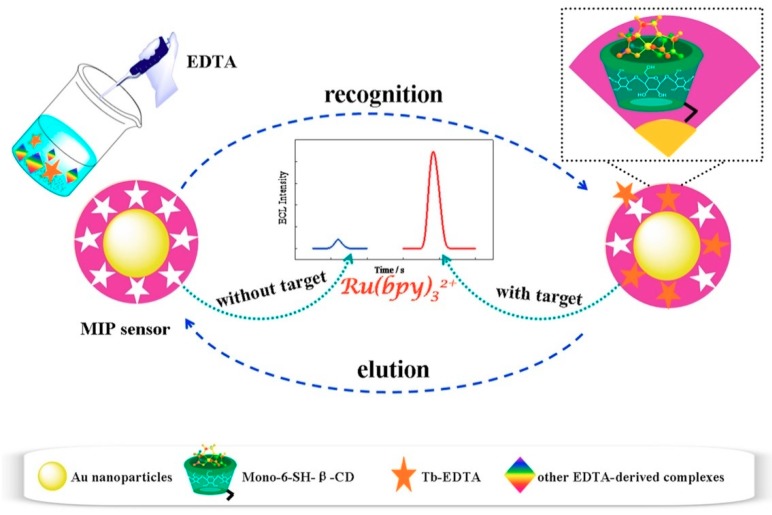
Scheme of the Russian Matryoshka structured molecularly imprinted sensor for ultra-trace Tb^3+^ determination as Tb-EDTA complex. Reproduced from [154] with permission of Elsevier.

**Figure 11 polymers-11-01173-f011:**
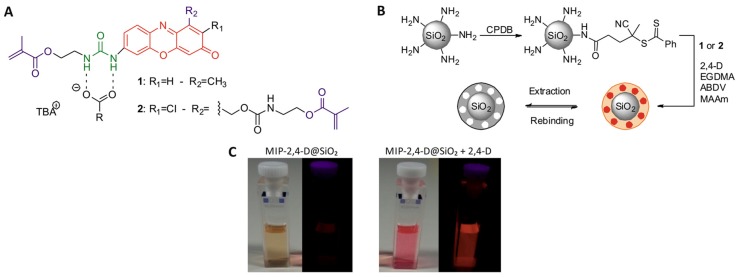
(**A**) Molecular architectures of the phenoxazinone monomer 1 and cross-linker 2. (**B**) Preparation of the core-shell microparticles from amine-modified silica microparticles with imprinting of the template into the MIP shell, followed by extraction and rebinding. (**C**) Photographs of 2 in CHCl_3_ in the absence (left) and presence (right) of 2,4-D/TBA in daylight and under UV lamp (365 nm) illumination. Reproduced from [164] with permission of Elsevier.

**Figure 12 polymers-11-01173-f012:**
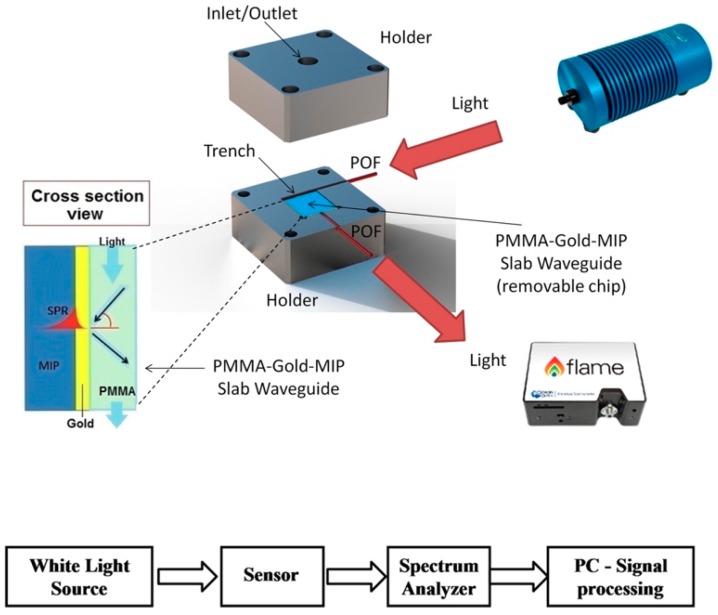
Schematic view of the SPR sensor selective to perfluorinated compounds based on a MIP receptor on the gold film placed in the middle of the holder connected by two plastic optical fibres. Top and cross section view of the sensor system and outline of the experimental setup depicting the instrumentation. Reproduced from Reference [186] with permission of Elsevier.

**Figure 13 polymers-11-01173-f013:**
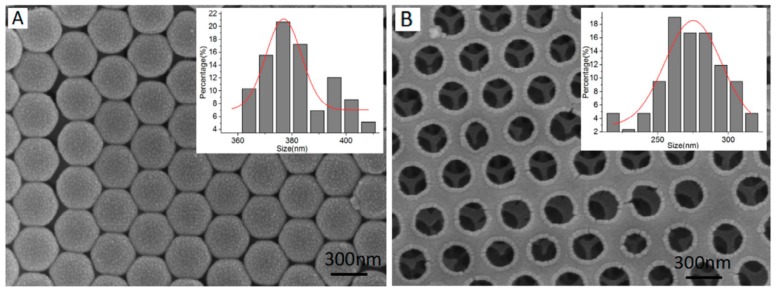
Scanning electron microscope (SEM) images of: (**A**) packed SiO_2_ arrays; (**B**) MIP-based inverse opal as a result of the infiltration of the pre-polymerization mixture, polymerization and chemical etching. Insets: particle size distributions. Reproduced from [197] with permission of Elsevier.

**Table 1 polymers-11-01173-t001:** Optical sensors based on MIPs deposited on the surface of a gold chip.

Composition	Polymerization	Detection	Analyte	LOD	Ref.
Au slide-MIP	GT Drop deposition	SPR–SPRi	Ciprofloxacin	0.08 μg L^−1^	[58]
Glass-Au nIsland-MIP	GT Spin-coating	LSPR	α-pinene	304 ppm	[60]
Au slide-MIP	GT Spin-coating	SPR	Anti-CCP	0.177 RU mL^−1^	[61]
Au slide-MIP film	GT Spin-coating	SPR	Amoxicillin	73 pM	[62]
Au slide-MIP	GF Polymerizable functionality	SPR	RoxP	0.23 nM	[64]
Au slide-MIP	GF Polymerizable functionality	SPR	*Salmonella paratyphi*	1.4 × 10^6^ CFU mL^−1^	[65]
Au slide-MIP	GF RAFT (Iniferter)	SPR	BSA	5.6 nM	[66]
Au slide/Ab/MIP	GF Monomer/Template immobilization	FL	Intact exosomes	6 pg mL^−1^	[67]
Au slide-β-CD/MIP	GF Monomer/Template immobilization	FL	Cortisol	4.8 pM	[68]
Au slide-MIP hydrogel	GF SI-ATRP	SPR	Lectin ConA	n. d.	[69]
Au slide-MIP	GF SI-ATRP	SPR	OVA	6.1 ng mL^−1^	[70]
Au slide-MIP	GT Electropolymerization	SPR	Histamine	2 μg mL^−1^	[71]
Au-MIP needles	GF Monomer/Template immobilization	SERS	Glucose/Fructose	1 μg mL^−1^	[72]
Au slide-nMIPs	nMIP immobilization	SPR	*Enterococcus faecalis*	1.05 × 10^2^ CFU mL^−1^	[73]
Au slide-nMIPs	nMIP immobilization	SPR	α-casein	127 ng mL^−1^	[74]
Au slide-nMIPs	nMIP immobilization	SPR	Vancomycin	4.1 ng mL^−1^	[75]
Glass-Au nanodisk-MIP	GF Monomer/Template immobilization	LSPR	Astringency *	n. d.	[76]
Au slide-MIP	GF Polymerizable functionality	SPR	Kanamycin	12 nM	[77]
Au slide-GR/MIP	GT In situ polymerization	SPR	L-tryptophan	0.105 mM	[78]
Glass-Au nIsland-MIP	GT Deposition polymerization	LSPR	VOCs	n.d.	[79]

* Expressed in pentagalloyl glucose (PGG) units. GF: Grafting from; GT: Grafting to; SI-ATRP: Surface-initiated atom transfer radical polymerization; RAFT: Reversible addition−fragmentation chain-transfer; LSPR: Localized surface plasmon resonance; SPR: Surface plasmon resonance; SPRi: Surface plasmon resonance imaging; SERS: Surface-enhanced Raman spectroscopy; FL: Fluorescence; Anti-CCP: Cyclic citrullinated peptide antibodies; ConA: Concanavalin A; BSA: Bovine serum albumin; VOCs: Volatile organic compounds; RoxP: Bacterial factor; OVA: Ovalbumin.

**Table 2 polymers-11-01173-t002:** Optical sensors based on MIPs deposited on the surface of a silicon-based slides.

Composition	Polymerization	Detection	Analyte	LOD	Ref.
GS–MIP	GT Deposition polymerization	IF	BSA	8.01 pg L^−1^	[59]
Quartz chip–MIP	GF Polymerizable functionality	FL	IBA	n. d.	[80]
GS–Zn porphyrin–MIP	GF Polymerizable functionality	FL	DMMP	0.1 μM	[81]
GS–MIP	MIP immobilization	SERS	Acid phosphatase	0.1 ng L^−1^	[82]
GS–nMIPs	nMIP immobilization	RIfS	PenG	4.32 mM	[83]
b-Si–Au–MIP	GT Deposition polymerization	SERS	Tetracycline	n.d.	[84]

GS: Glass slide; nMIP: nano MIP; GF: Grafting from; GT: Grafting to; DMMP: Dimethyl Methylphosphonate; SERS: Surface-enhanced Raman spectroscopy; FL: Fluorescence; IF: Interferometry; RIfS: Reflectometric Interference Spectroscopy; BSA: Bovine serum albumin; IBA: Indole-3-Butyric Acid; PenG: Penicillin G.

**Table 3 polymers-11-01173-t003:** Optical sensors based on MIPs deposited on the surface of an electrode.

Composition	Polymerization	Detection	Analyte	LOD	Ref.
CPE–MIP	Mixed MIP-CP	ECL	Azithromycin	23 pM	[87]
GCE-MIP	GT Electropolymerization	ECL	*E. coli* O157:H7	8 CFU mL^−1^	[88]
GCE–rGO–UCNPs-MIP	GT Electropolymerization	ECL	Clenbuterol	6.3 nM	[89]
GCE-CNT-AuNP-MIP	GT Electropolymerization	ECL	Triazophos	3.1 ng L^−1^	[90]
Au/[Ru(bpy)_3_]^2+^/MWCNTs/nTiO_2_-MIP	MIP Deposition	ECL	Bisphenol A	4.1 ng L^−1^	[91]
GCE-RUDS/MIP	MIP drop-cast	ECL	Melamine	500 fM	[92]
GCE-Ru@ethyl-SiO2-MIP	GT Electropolymerization	ECL	17ß-Estradiol	5 pg L^−1^	[93]
GCE–AuNPs-Ru@SiO_2_-MIP	GT Drop deposition	ECL	Fumonisin B1	0.35 pg mL^−1^	[94]
GCE-CdTe-Ru@SiO_2_-MIP	GT Drop deposition	ECL	α-ergocryptine OTA	0.18 fg mL^−1^ 0.25 fg mL^−1^	[95]
GCE–GO-Au–Aptamer-CDs/MIP	GT Electropolymerization	ECL	Lincomycin	160 fM	[96]
ITO–GO/CDs-MIP	GT Electropolymerization	FL	Virginiamycin	15.6 pM	[97]
ITO-IrOx NPs–MIP	GT Drop deposition	EC	Chlorpyrifos	0.1 pM	[98]
ITO-GO-Fe_3_O_4_/NiNCs/MIP	GF Polymerizable functionality	ECL	Creatinine	0.5 nM	[99]
GE–QD-MIP	GT Electropolymerization	ECL	Clopyralid	4.1 pM	[100]
GCE–AuNPs/GO–MIP	GF Monomer/Template immobilization	ECL	Alpha-fetoprotein	0.4 ng L^−1^	[101]

MWCNTs: Multi-walled carbon nanotubes; CNT: Carbon nanotubes; AuNP: Gold nanoparticles; GCE: Glassy carbon electrode; RUDS: Silica nanoparticles doped with [Ru(bpy)_3_]^2+^; GO: Graphene oxide; CDs: Carbon dots; UCNPs: Up-conversion nanoparticles; ITO: Indium Tin Oxide; IrOx: Iridium oxide; GF: Grafting from; GT: Grafting to; NiNCs: Nickel nanoclusters; CPE: Carbon paste electrode; QDs: Quantum dots; GE: Gold electrode; ECL: Electrochemiluminescence; EC: Electrochromism; FL: Fluorescence; OTA: Ochratoxin A.

**Table 4 polymers-11-01173-t004:** Optical sensors based on core-shell MIPs with a metal-oxide core.

Composition	Polymerization	Detection	Analyte	LOD	Ref.
ZnO NRs@MIP	GT Coating	FL	Tetracycline	1.27 μM	[103]
ZnO/Ag NRs@MIPs	GF Polymerizable functionality	SERS	R6G	10 pM	[105]
MoO_3_ NRs@MIP	GF Polymerizable functionality	SERS	Methylene blue	0.16 mM	[106]
ZnO NRs@MIPs	GF Polymerizable functionality	FL	DEHP	1.83 nM	[107]

NRs: Nanorods; GF: Grafting from; GT: Grafting to; FL: Fluorescence; SERS: Surface-enhanced Raman scattering; R6G: Rhodamine 6G; DEHP: Diethylhexyl phthalate.

**Table 5 polymers-11-01173-t005:** Optical sensors based on core-shell MIPs with a magnetic core.

Composition	Polymerization	Detection	Analyte	LOD	Ref.
Fe_3_O_4_@MIP	GT Coating	RLS	HAV	6.2 pM	[109]
Fe_3_O_4_@SiO_2_@MIP	GF Polymerizable functionality	FL	Rhodamine B	0.1 nM	[110]
Fe_3_O_4_@Au@MIP	GT Coating	ECL	Cinchonine	0.313 pM	[111]
Fe_3_O_4_@MIP/QDs	GT Coating	FL	NDPhA	0.69 μM	[112]
Fe_3_O_4_@SiO_2_@MIP	GT Coating	ECL	Diethylstilbestrol	0.1 pg mL^−1^	[113]
Fe_3_O_4_@SiO_2_@QDs@MIP	GF Polymerizable functionality	FL	Dibutyl phthalate	80 nM	[114]

QDs: Quantum dots; GF: Grafting from; GT: Grafting to; ECL: Electrochemiluminescence; FL: Fluorescence; RLS: Resonance light scattering; HAV: Hepatitis A virus; NDPhA: *N*-Nitrosodiphenylamine.

**Table 6 polymers-11-01173-t006:** Optical sensors based on core-shell MIPs with an up-conversion nanoparticles core.

Composition	Polymerization	Detection	Analyte	LOD	Ref.
UCNPs@Apta-MIP	GT Coating	FL	Enrofloxacin	0.04 ng mL^−1^	[116]
UCNPs@MIPs–AgNPs	GT Coating	FL SERS	Histamine	9 μg L^−1^	[117]
UCNPs@SiO_2_@MIP	GF Polymerizable functionality	FL	Diethylstilbestrol	12.8 ng mL^−1^	[118]
UCP@Fe_3_O_4_@MIP	GT Coating	FL	Enrofloxacin Ciprofloxacin Enoxacin Fleroxacin Levofloxacin	2.50 × 10^−7^ M 1.24 × 10^−6^ M 1.12 × 10^−6^ M 4.06 × 10^−7^ M 1.47 × 10^−6^ M	[119]

UCNPs: Up-conversion nanoparticles; AgNPs: Silver nanoparticles; UCPs: Up-conversion particles; GF: Grafting from; GT: Grafting to; FL: Fluorescence; SERS: Surface-enhanced Raman spectroscopy.

**Table 7 polymers-11-01173-t007:** Optical sensors based on core-shell MIPs with a quantum dot core.

Composition	Polymerization	Detection	Analyte	LOD	Ref.
CdSe/ZnS@SiO_2_@MIP	GT Coated	FL	DMHF C4-HSL C6-HSL C8-HSL *N*-3oxo-C6-HSL	0.66 nM 0.54 nM 0.88 nM 0.72 nM 0.68 nM	[121]
CdTe@MIP	GT Coated	RFL	BHb	9.6 nM	[122]
FeSe@MIP	GT Coated	FL	Cyfluthrin	1.0 μg kg^−1^	[123]
Mn:ZnS@MIP	GT Entrapped	FL	Sulfapyridine	0.5 μM	[124]
CsPbBr_3_@MIP	GT Coated	FL	Omethoate	18.8 μg L^−1^	[125]
CDs@MIP	GT Coated	FL	R6G	n.d.	[126]
GDs@MIP	GT Entrapped	FL	Ornidazole	0.24 μM	[127]
CdTe@SiO_2_@MIP	GT Coated	FL	Neomycin	0.16 μg L^−1^	[128]
MIP@CdSe	GF Polymerizable functionality	FL	Kanamycin	13 μg L^−1^	[129]
Gra-CdSe/ZnS@MIP	GT Entrapped	FL	Tyramine	21 μg L^−1^	[130]
CdSe@SiO_2_/CD/MIP	GT Coated	RFL	4-nitrophenol	26 ng L^−1^	[131]
CDs@SiO_2_@MIP/CdTe	GT Coated	RFL	Celecoxib	57 nM	[132]
CdTe/ZnQ_2_@mMIP	GT Coated	RFL	Brilliant Blue	8.8 nM	[133]
GDs@MIPCdTe@MIP	GF Polymerizable functionality	FL	Cephalexin Ceftriaxone	0.06 μg L^−1^ 0.10 μg L^−1^	[134]
CDs@MIP	GT Coated	FL	Phenobarbital	0.1 nM	[135]
CdTe@MIP	GT Coated	RFL	Phycocyanin	3.2 nM	[136]
CDs@SiO_2_@MIP/CdTe	GT Coated	RFL	Sulfadiazine	0.042 μM	[137]
CdSe/ZnS@MIP	GF Polymerizable functionality	FL	Acetaminophen	0.34 nM	[138]
Mn:ZnS @MIP	GF Polymerizable functionality	FL	Cyt C	0.054 μM	[139]
MIP@CDs	GT Entrapped	FL	3-nitrotyrosine	17 nM	[140]
MIP@CDs	GT Coated	FL	PrHy	0.5 μM	[141]
MIP@CDs	GT Coated	FL	Tetracycline	9 nM	[142]
Mn:ZnS@MIP	GT Coated	FL	Cocaine	15–35 μg L^−1^	[143]
Mn:ZnS@SiO_2_@MIPs	GT Entrapped	FL	Serotonin	0.69 μg L^−1^	[144]
CDs@MIP	GT Coated	FL	3-MCPD	0.6 μg L^−1^	[145]
CdTe@SiO_2_@MIP	GF Polymerizable functionality	FL	TBBPA	0.3 μg L^−1^	[146]
ZnO@MIP	GT Coated	FL	THZ	0.43 nM	[147]
CDs@MIP	GT Entrapped	FL	Acetamiprid	2 nM	[148]
CdTe/CdS@MIP	GT Entrapped	FL	PFOA	25 nM	[149]
CdTe@MIPs	GF Polymerizable functionality	FL	*p*-coumaric acid	6.74 ng L^−1^	[150]

CDs: Carbon dots; Mn:ZnS: Mn-doped ZnS; GDs: Graphene dots; CdTe/ZnQ_2_: CdTe/ZnS quelated by 8-hydroxyquinoline; mMIP: Mesoporous MIP; GF: Grafting from; GT: Grafting to; BHb: Bovine haemoglobin; 4-NP: 4-nitrophenol; TNP: 2,4,6-trinitrophenol; C4-HSL: *N*-butyryl-L-homoserine lactone: C6-HSL: *N*-hexanoyl-L-homoserine lactone; C8-HSL: *N*-octanoyl-L-homoserine; *N*-3oxo-C6-HSL: 3-oxo-hexanoyl homoserine lactone; R6G: Rhodamine 6G; Cyt C: cytochrome C; PrHy: Promethazine hydrochloride; 3-MCPD: 3-Monochloropropane-1,2-diol; TBBPA: Tetrabromobisphenol-A; THZ: Thioridazine hydrochloride; PFOA: Perfluorooctanoic acid; FL: Fluorescence; RFL: Ratiometric fluorescence.

**Table 8 polymers-11-01173-t008:** Optical sensors based on MIPs deposited on/grown from a metal core.

Composition	Polymerization	Detection	Analyte	LOD	Ref.
AuNPs@nanoNIPs	GT Embedding	SERS	Sudan IV	n.d.	[152]
bAuNPs@mSiO_2_@MIP	GF RAFT	SERS	Enrofloxacin	1.5 nM	[153]
AuNP@CD/MIP	GT Coating	ECL	Tb^3+^	39 pM	[154]
AgNPs@MIP (different shapes)	GF Initiator immobilization (ATRP)	PL	Phenformin	*	[155]
Ag@QDs@MIPs	GT ATRP	SERS	2,6-DCP	1 μM	[156]
AuNPs/MIP–SPR Chip	GT Embedding	SPR	RDX	10 fM	[157]
Ag@MIP	GF Polymerizable functionality	SERS	Cyhalothrin	13 nM	[158]
MIPs/AgNPs	GT Embedding	SERS	Bisphenol A	0.5 μM	[159]

* Detection limits depend on the shape of the nanoparticles and the detection system used. The reader is referred to the corresponding work for further information. bAuNPs: Multibranched gold nanoparticles; mSiO_2_: Mesoporous silica; CD: β-cyclodextrin; GF: Grafting from; GT: Grafting to; RAFT: Reversible addition−fragmentation chain-transfer; ATRP: Atom transfer radical polymerization; SERS: Surface-enhanced Raman scattering; SPR: Surface plasmon resonance; PL: Photoluminescence; ECL: Electrochemiluminescence; RDX: 1,3,5-trinitroperhydro-1,3,5-triazine; 2,6-DCP: 2,6-dichlorophenol; n.d.: Not described.

**Table 9 polymers-11-01173-t009:** Optical sensors based on MIPs deposited on a silica core.

Composition	Polymerization	Detection	Analyte	LOD	Ref.
SiO2@FMIPs	GT Coating	FL	JEV	9.6 pM	[160]
SiO_2_@MIP	GT Coating	RLS	HAV	8.6 pM	[161]
SiO_2_@MIP	GF Polymerizable functionality	RLS	OVA	0.13 nM	[162]
SiO_2_@FMIPs	GF Polymerizable functionality	FL	τ-fluvalinate	12.14 nM	[163]
SiO_2_@FMIP	GF RAFT	FL	2,4-D	28 nM	[164]
SiO_2_@Ag@MIPs	GF Polymerizable functionality	SERS	R6G	1 pM	[165]
SiO2@AgNPs@MIP	GT Coating	SERS	Bisphenol A	0.146 pM	[166]
SiO_2_@AuNPs@MIPs	GT Coating	FL	Haemoglobin	0.03 μM	[167]
SiO_2_@QDs@MIP	GF ATRP	FL	λ-cyhalothrin	0.13 μM	[168]
SiO_2_@QDs@MIP	GF Polymerizable functionality	FL	Dibutyl phthalate	0.04 μM	[169]
SiO_2_@QDs@m-MIP	GT Coating	FL	2,4-D	2.1 nM	[170]
SiO_2_@FMIP	GT Coating	FL	τ-fluvalinate	13.251 nM	[171]
SiO_2_@QDs@m-MIP	GT Coating	RTP	Transferrin	14 nM	[172]

FMIPs: Fluorescent MIPs; AuNPs: Gold nanoparticles; QDs: Quantum dots; m-MIP: Mesoporous MIP; AgNPs: Silver nanoparticles; GF: Grafting from; GT: Grafting to; ATRP: Atom transfer radical polymerization; RAFT: Reversible addition−fragmentation chain-transfer; FL: Fluorescence; RLS: Resonance light scattering; SERS: Surface-enhanced Raman scattering; RTP: Room-temperature phosphorescence; JEV: Japanese encephalitis virus; HAV: Hepatitis A virus; R6G: Rhodamine 6G; OVA: Ovalbumin; 2,4-D: 2,4-dichlorophenoxyacetic acid.

**Table 10 polymers-11-01173-t010:** Optical sensors based on MIPs deposited on an optical fibre.

Composition	Polymerization	Detection	Analyte	LOD	Ref.
POF-Gold-MIP	GT Deposition	SPR	DBDS Furfural	0.346 μM 0.048 μM	[183]
Double POF-MIP	GT Deposition	SPR	DBDS	50 μM	[184]
Double POF-MIP	GT Deposition	SPR	DBDS	53 μM	[185]
POF-Gold-MIP	GT Deposition	SPR	Furfural	0.03 ppm	[186]
POF–MIP-Ag	GT Coating	LSPR + SPR	Ascorbic acid	0.74 pM	[187]
POF-ZnO/MoS_2_-MIP		LMR	*p*-cresol	28 nM	[188]
POF-Gold-MIP	GT Deposition	SPR	Perfluorinated Compounds	0.13 ppb	[189]

POF: Plastic optical fibre; GT: Grafting to; GF: Grafting from; SPR: Surface plasmon resonance; LSPR: Localized surface plasmon resonance; LMR: Lossy mode resonance; DBDS: Dibenzyl disulphide.

**Table 11 polymers-11-01173-t011:** Optical sensors based on photonic MIPs.

Composition	Detection	Analyte	LOD	Ref.
PC-MIP	Red shift Reflection	Sulfaguanidine	0.28 nM	[197]
MIPPs	Blue-shift Diffraction	Sulfonamides	3.8 µM *	[198]
Imprinted RIOPs	FL	Enrofloxacin	0.082 ppb	[199]
Au-MIP IO PCs	Red shift	Parathion	1 ng L^−1^ *	[200]
PC-MIP	Colorimetric array Diffraction	TNT2, 6-DNT2, 4-DNT 4-MNT	3.53 µg 2.42 µg 4.85 µg 2.14 µg	[201]
MIPH	Red shift Diffraction	L-histidine	10 pM	[202]
MIP IO spheres	Red shift Reflection	MPA	1 µM *	[203]
CC–MIP-CDs	FL	2,4-DNT	1 mM *	[204]

* Lowest concentration quantified. PC: Photonic crystal; MIPPs: Molecularly imprinted photonic polymers; RIOPs: Responsive inverse opal polymers; IO PCs: Inverse opal photonic crystals; MIPH: Molecularly imprinted photonic hydrogel; CC: Colloid crystal; CDs: Carbon dots; FL: Fluorescence; TNT: 2,4,6-trinitrotoluene; 2,6-DNT: 2,6-dinitrotoluene; 2,4-DNT: 2,4-dinitrotoluene; 4-MNT: 4-nitrotoluene; MPA: Methanephosphonic acid.

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
