# Peer review of "Molecularly Imprinted Polymer-Based Hybrid Materials for the Development of Optical Sensors"

_polymers, 2019, doi:10.3390/polym11071173_

Reviewer 1 Report

polymers-541762

The manuscript is a review describing recent developments of MIP based hybrid materials for its use with optical sensors. I have found the review interesting and informative. The manuscript is well-written and has a clear structure. Tables and figures help to give an organized overview of the subject. I have a few suggestions for improving the manuscript, but with these changes, the document should be published.

Fig. 11 is not visible in my version.

The whole manuscript does not contain information regarding the reproducibility. Are information available? The most sensors are not commercially produced and are home-made in the labs. Information of the reproducibility between these different sensors would be helpful.

Author Response

The manuscript is a review describing recent developments of MIP based hybrid materials for its use with optical sensors. I have found the review interesting and informative. The manuscript is well-written and has a clear structure. Tables and figures help to give an organized overview of the subject. I have a few suggestions for improving the manuscript, but with these changes, the document should be published.

A: We thank the referee for the comments.

Fig. 11 is not visible in my version.

A: The figure has modified and inserted again.

The whole manuscript does not contain information regarding the reproducibility. Are information available? The most sensors are not commercially produced and are home-made in the labs. Information of the reproducibility between these different sensors would be helpful.

A: Generally, in this kind of studies, reproducibility is evaluated at the level of sensor regeneration. We have added now a few sentences when this studies are performed:

•             Page 9 – Lines 330-332.

•             Page 10 – Lines 347-348.

•             Page 11 – Lines 398-400.

•             Page 12 – Lines 430-431.

•             Page 15 – Lines 529-531.

•             Page 18 – Lines 606-607 and 613-614.

•             Page 19 – Lines 649-650.

•             Page 22 – Lines 699-700.

Reviewer 2 Report

This paper reviews molecularly imprinted polymer-based hybrid materials for the development of optical sensors. The review was well written. The reviewer only has two questions. First, it is better to add a table of content ahead of Introduction. Second, the organization and couple of subtitles of the manuscript can be improved. For example, part 2 -  bidimensional composites focuses on the description of techniques and approaches to manufacture MIP-based sensors. Is the "Bidimensional composites" one of these techniques or the name of summarizing all the techniques? Perhaps just use "manufacturing techniques and approaches of MIP-based hybrid materials" instead of "bidimensional composites". The subtitle of part 3 core-shell formats may be the same problem which generates a confusion and might not fully cover the content described in part 3.     

Author Response

This paper reviews molecularly imprinted polymer-based hybrid materials for the development of optical sensors. The review was well written.

A: We thank the referee for the comments.

The reviewer only has two questions. First, it is better to add a table of content ahead of Introduction.

A: A table of content has been added according to the reviewer’s recommendations.

Second, the organization and couple of subtitles of the manuscript can be improved. For example, part 2 -  bidimensional composites focuses on the description of techniques and approaches to manufacture MIP-based sensors. Is the "Bidimensional composites" one of these techniques or the name of summarizing all the techniques? Perhaps just use "manufacturing techniques and approaches of MIP-based hybrid materials" instead of "bidimensional composites". The subtitle of part 3 core-shell formats may be the same problem which generates a confusion and might not fully cover the content described in part 3.    

A: We wanted to distribute the manuscript according to the dimensionality of the composite obtained. However, we agree with the referee that this may be confusing to the reader as the subheading section “bidimensional hybrids”, in the form of layers onto substrates, may be different from the one including core-shell materials. Thus, we have decided to keep the heading of section 2 under the name of “bi-dimensional composites”, while the name of section 3 has been replaced by “three-dimensional composites” to avoid this confusion.
